# On the Role of Attention Masks and LayerNorm in Transformers

**Xinyi Wu**[1]     **Amir Ajorlou**[1]     **Yifei Wang**[2]     **Stefanie Jegelka**[3,2]     **Ali Jadbabaie**[1]

[1]MIT LIDS     [2]MIT CSAIL     [3]TU Munich

{xinyiwu,ajorlou,yifei_w,stefje,jadbabai}@mit.edu

## Abstract

Self-attention is the key mechanism of transformers, which are the essential building blocks of modern foundation models. Recent studies have shown that pure self-attention suffers from an increasing degree of rank collapse as depth increases, limiting model expressivity and further utilization of model depth. The existing literature on rank collapse, however, has mostly overlooked other critical components in transformers that may alleviate the rank collapse issue. In this paper, we provide a general analysis of rank collapse under self-attention, taking into account the effects of attention masks and layer normalization (LayerNorm). In particular, we find that although pure masked attention still suffers from exponential collapse to a rank one subspace, sparse or local masked attention can provably slow down the collapse rate. In the case of self-attention with LayerNorm, we first show that for certain classes of value matrices, collapse to a rank one subspace still happens exponentially. However, through construction of nontrivial counterexamples, we then establish that with proper choice of value matrices, a general class of sequences may not converge to a rank one subspace, and the self-attention dynamics with LayerNorm can simultaneously possess a rich set of equilibria with any possible rank between one and full. Our result refutes the previous hypothesis that LayerNorm plays no role in the rank collapse of self-attention and suggests that self-attention with LayerNorm constitutes a much more expressive, versatile nonlinear dynamical system than what was originally thought.

## 1 Introduction

The celebrated attention mechanism has proven highly effective in the architecture of transformers [35], which serve as the key building block of modern foundation models including large language models. From a theoretical perspective, understanding the underlying mechanism of transformers and attention in general has become pivotal for elucidating existing models and paving the way for developing more powerful future models [12, 15].

Transformers are known to exhibit the *rank collapse phenomenon*[1] [12, 15, 20, 27, 33, 37], which refers to the observation that increasing the number of self-attention layers leads to homogeneous token representations. To gain more insight into the rank collapse issue and better understand the effects of self-attention in multi-layer models, it is essential to study the long-term behavior of tokens under self-attention dynamics [12, 15, 16].

However, many existing studies do not take into account architectural components that are commonly used in practice. For instance, first, the theoretical analysis of long-term self-attention dynamics often assumes that attention is *fully bidirectional* — that is, all tokens are allowed to attend to all other tokens in the sequence [12, 15, 16, 27], which only applies to certain attention mechanisms

---

[1]This is also referred to as the *oversmoothing* [15, 33, 37] or the *token uniformity* [12, 27] problem.

38th Conference on Neural Information Processing Systems (NeurIPS 2024).

such as the one deployed in the BERT family [11, 24]. The vast majority of popular transformer architectures used nowadays in language models, including the GPT models [6, 30], use the causal attention masks where tokens are only allowed to attend to preceding tokens, or have sparse attention structure (sparse attention) [4, 8, 10, 21, 22, 32, 38, 41, 42], where attention is restricted to local interactions between tokens and their chosen neighbors. This limits the practical applicability of the theoretical results developed in [12, 15, 16, 27], as they heavily rely on the key assumption that attention is fully bidirectional.

Second, layer normalization (LayerNorm) is another inherent component of transformers. Dong et al. [12] put forth a hypothesis that LayerNorm plays no role in preventing rank collapse in self-attention networks. This hypothesis is then partially validated by a continuous-time analysis of self-attention dynamics conducted in Geshkovski et al. [15]. The analysis, which incorporates LayerNorm, shows the exponential convergence of tokens to a common point on the unit sphere. However, this result relies on a strong assumption that all the value matrices are the identity matrix. In contrast, for multilayer perceptrons (MLPs), Joudaki et al. [23] show that LayerNorm improves the isometry of the representations and can prevent rank collapse in that setting. It thus remains a question whether it is truly the case that LayerNorm could not have a similar effect in transformers under more general assumptions due to self-attention.

Hence, in this work, we rigorously study the effect of attention masks and LayerNorm on the token dynamics and rank collapse in transformers. The main questions we address are as follows:

> *Can attention masks alleviate rank collapse of tokens under self-attention? If so, what type of attention mask would be the most effective?*

> *Can LayerNorm alleviate rank collapse of tokens under self-attention? If so, what long-term behavior of tokens would LayerNorm lead to?*

We answer these questions through a rigorous analysis of the self-attention dynamics. Notably, unlike some previous works [15, 16], which regard self-attention as a continuous-time dynamical system, we view self-attention as a discrete-time dynamical system, more closely resembling the architecture used in practice. Furthermore, our analysis extends the results to more general masked attention by making novel use of a graph-theoretic approach and incorporating advanced results on infinite products of inhomogeneous non-negative matrices that may be of independent interest.

**In summary, we make the following contributions:**

- We establish that with pure self-attention, the exponential convergence of tokens to a common representation holds for a broad class of attention masks, accounting for the causal mask and a wide class of sparse attention patterns such as the sliding window [4, 42]. The key property that leads to the exponential rank collapse is a token that serves as a common "context" for all the other tokens in the sequence to directly or indirectly attend to. Our results also show that *local* attention can slow down the rate of rank collapse, suggesting its potential advantage over full bidirectional attention from an expressivity perspective at finite depth.

- We show that with LayerNorm, when the value matrices are orthogonal, the exponential convergence of tokens to a common point on the unit sphere holds for a broad class of attention masks. Nonetheless, by constructing nontrivial counterexamples, we prove that the self-attention dynamics with LayerNorm can simultaneously have a rich set of equilibria of any possible rank ranging from one to full. Moreover, we rigorously establish that self-attention with LayerNorm, together with proper choice of value matrices, can provably prevent complete collapse of tokens to a rank one subspace for a generic class of input sequences.

## 2   Related work

**Analysis of self-attention dynamics**   Understanding the attention mechanism is a pivotal step towards understanding the inner workings of transformer-based models. From a dynamical systems perspective, one could abstract the forward pass of the model as tokens undergoing a nonlinear, time-varying dynamics determined by the self-attention mechanism and other parameters of the model. Specifically, Dong et al. [12] first show that as the number of self-attention layers increases, tokens inevitably suffer from exponential rank collapse, while Wu et al. [37] establish a similar result for the attention mechanism in graph neural networks, taking additionally the layer-wise nonlinear

activation functions into account. Other works [15, 16] take a continuous-time approach and study more fine-grained clustering behaviors of the dynamics in transformers.

**The effect of LayerNorm in transformers**   LayerNorm is one of the most commonly used normalization techniques in modern neural networks [3] and has become an inherent component of transformers [35]. To better understand its role in transformers, Xiong et al. [39] study it from an optimization perspective and show that LayerNorm stabilizes the gradients during the backward pass. In addition, Brody et al. [5] find that LayerNorm plays a crucial role in improving the expressivity of the attention layer by making it easier for the model to compute the most frequent token in the input and avoid the problem of "unselectable" keys. Most relevant to our work, Dong et al. [12] mention the effect of LayerNorm in terms of mitigating the rank collapse issue in transformers. The paper makes a hypothesis that LayerNorm has no effect for token rank collapse. The argument is based on a heuristic observation (see Appendix A for a detailed discussion) which is at odds with the case of simpler models such as MLPs where LayerNorm is shown to be pivotal in addressing a similar rank collapse problem [23]. It thus remains an open question what effect LayerNorm would have on rank collapse in transformers.

**Sparse and local Attention**   While many existing transformer models and LLMs do not use sparse attention, sparse and local attention is gaining popularity due to the demand for efficiency, particular for long-context tasks. For example, sparse attention was populated by Longformer [4] and OpenAI [10] and nowadays popular LLMs like Mistral 7B [22] use sliding window attention by default. Other popular sparse attention models include, but are not limited to BigBird [42], Recurrent Memory Transformers (RMTs) [7], and Streaming Attention [38]. Besides language tasks, sparse attention is also common in vision transformers [19, 25, 28].

## 3   Problem Setup

**Notation**   Let $\|\cdot\|_2, \|\cdot\|_F$ be the 2-norm and Frobenius norm, respectively. We use the shorthand $[n] := \{1, \ldots, n\}$. We denote the all-one vector of length $N$ by $\mathbf{1} \in \mathbb{R}^N$. For a matrix $M$, we denote its $i$-th row by $M_{i,:}$ and its $j$-th column by $M_{:,j}$.

Throughout the analysis in the paper, we formalize the attention mask to be a directed graph $\mathcal{G}$. Formally, we represent a directed graph with $N$ nodes by $\mathcal{G}$ and let $E(\mathcal{G})$ be the set of directed edges of $\mathcal{G}$. If there is a directed edge going from $j$ to $i$ in $\mathcal{G}$, i.e. $(j, i) \in E(\mathcal{G})$, for the attention mechanism it means that token $j$ serves as a direct context for token $i$ or token $i$ attends to token $j$. The set $\mathcal{N}_i$ of all neighbors of node $i$ is then $\{k : (k, i) \in E(\mathcal{G})\}$.

Furthermore, we will be using the following graph-theoretic terminology:

**Definition 1** (Reachability). *We say a node $v$ is* reachable *from $u$ in a directed graph $\mathcal{G}$ if and only if there is a directed path $(u, n_1), (n_1, n_2), ..., (n_k, v)$ from $u$ to $v$.*

**Definition 2** (Strongly Connected). *A directed graph $\mathcal{G}$ is said to be* strongly connected *if and only if any two distinct of nodes are reachable from each other.*

**Definition 3** (Center Node). *A node $v$ from which every node in the directed graph $\mathcal{G}$ is reachable is called a* center node.

**Definition 4** (Quasi-Strongly Connected). *A directed graph $\mathcal{G}$ is said to be* quasi-strongly connected *if $\mathcal{G}$ has at least one center node.*

**Definition 5** (Radius). *The radius of a quasi-strongly connected graph is defined to be the longest distance from a center node to any node in the graph. If there are multiple center nodes, then it is the smallest value among them.*

### 3.1   (Masked) Attention Mechanism

Given token representations $X \in \mathbb{R}^{N \times d}$, the raw attention score matrix is computed as

$$R = XW_Q(XW_K)^\top / \sqrt{d_{QK}},$$

where $W_Q, W_K \in \mathbb{R}^{d \times d'}$ are the query and the key matrix, respectively, and $\sqrt{d_{QK}}$ is a temperature term to control the scale of raw attention scores. To enforce a masked attention, we create a sparse

attention matrix $A \in \mathbb{R}^{N \times N}$ based on $R$ whose sparsity pattern is specified by a directed graph $\mathcal{G}$: we normalize $R_{ij}$ among all allowed token attention interactions $(k, i) \in E(\mathcal{G})$ such that

$$A_{ij} = \mathrm{softmax}_{\mathcal{G}}(R_{ij}) = \frac{\exp(R_{ij})}{\sum_{k \in \mathcal{N}_i} \exp(R_{ik})} \text{ if } (j, i) \in E(\mathcal{G}), \quad A_{ij} = 0 \text{ otherwise.}$$

## 3.2 LayerNorm

Given token representations $X \in \mathbb{R}^{N \times d}$, LayerNorm subtracts the mean across different columns in each row and then scales each row to have a unit 2-norm. In this work, we consider LayerNorm to only perform the scaling operation, which is a common assumption in theoretical analyses of the attention mechanism [15, 34][2]. Mathematically, let $D = \mathrm{diag}(d_1, d_2, ..., d_N)$ where $d_i = 1/\|X_{i,:}\|_2$ for all $i \in [N]$, then $\mathrm{LN}(X) = DX$ .

## 3.3 Self-attention dynamics

For our analysis, we consider single-head (masked) self-attention networks (SANs), where the layerwise update rule can be written as

$$A^{(t)} = \mathrm{softmax}_{\mathcal{G}^{(t)}} \left( X^{(t)} W_Q^{(t)} (X^{(t)} W_K^{(t)})^\top / \sqrt{d_{QK}} \right) \tag{1}$$

$$\tilde{X}^{(t+1)} = A^{(t)} X^{(t)} W_V^{(t)} \tag{2}$$

$$X^{(t+1)} = \mathrm{LN}(\tilde{X}^{(t)}) := D^{(t)} A^{(t)} X^{(t)} W_V^{(t)} . \tag{3}$$

where $W_V^{(t)} \in \mathbb{R}^{d \times d'}$ is the value matrix. For simplicity, throughout the paper, we assume that $d = d'$ and $\mathcal{G}^{(t)} = \mathcal{G}$, i.e. the attention mask is the same for all the attention layers. Yet the results can be easily generalized to the case where masks are time-varying and satisfy similar regularity conditions.

## 4 Main Results: Attention with Masking and LayerNorm

To study how token representations evolve under the self-attention dynamics and behave in the long-term, we measure token similarity via $\mu(\cdot) : \mathbb{R}^{N \times d} \to \mathbb{R}_{\geq 0}$:

$$\mu(X) := \|X - \mathbf{1}\gamma_X\|_F, \text{ where } \gamma_X = \mathbf{1}^\top X / N . \tag{4}$$

This measure is mathematically equivalent to the measure $\mathrm{res}(X) = \arg\min_{x \in \mathbb{R}^d} \|X - \mathbf{1}x^\top\|_F$ used in [12], but the form in (4) is easier to work with in the general analysis and more direct to compute. Another advantage of our formulation is that it clearly demonstrates that Theorem 1 and Theorem 2 are not dependent on the specific choice of $\mu(\cdot)$: these results apply to any Lipschitz $\mu'(\cdot)$ with a Lipschitz constant $L$ such that $\mu(X) = 0$ if and only $X_{i,:} = X_{j,:}, \forall i, j \in [N]$, as we can use the formulation to directly derive that

$$\mu'(X) = |\mu'(X) - \mu'(\mathbf{1}_{\gamma_X})| \leq L\|X - \mathbf{1}_{\gamma_X}\|_F = L\mu(X) .$$

Finally, we adopt the following assumptions in our analysis:

**A1** $\mathcal{G}$ contains self-loops. i.e. $(i, i) \in E$ for every token $i \in [N]$ .

**A2** There exist constants $C \in \mathbb{R}$ such that $\max_{t \in \mathbb{N}} \left\{ \left\|W_Q^{(t)}\right\|_2, \left\|W_K^{(t)}\right\|_2 \right\} \leq C.$

**A1** ensures that every token has a neighbor so that the masked attention computation is well-defined for every token in every layer, while **A2** assumes that the key and query weight matrices are bounded, which is key for efficient attention computation in practice [2].

---

[2]This definition of LayerNorm, precisely concides with the RMSNorm [43] that is widely deployed in mainstream LLMs. We choose the terminology following the convention of literature.

## 4.1 Masked Attention

We first analyze the case without LayerNorm and focus on the effect of the attention mask. To ensure boundedness of the token trajectories $X^{(t)}$ for all $t \geq 0$ even without LayerNorm, we further assume that

**A3** The sequence $\left\{ \left\| \prod_{t=0}^{k} W_V^{(t)} \right\|_2 \right\}_{k=0}^{\infty}$ is bounded.

Then with general attention masks $\mathcal{G}$, there remains a strong connection between tokens via attention, and the token representations collapse exponentially to rank one.

**Theorem 1.** *Consider the self-attention dynamics without LayerNorm defined in* (2). *Under* **A1**-**A3**, *if* $\mathcal{G}$ *is a quasi-strongly connected graph, then there exists* $\epsilon > 0$ *where for all* $t \geq 0$,

$$A_{i,j}^{(t)} \geq \epsilon, \quad \text{for all } (j, i) \in E. \tag{5}$$

*As a result, a rank collapse of tokens happens exponentially with respect to* $\mu(\cdot)$, *i.e. there exists* $C > 0$ *such that*

$$\mu(X^{(t)}) \leq C \left(1 - \epsilon^r\right)^{t/r}, \tag{6}$$

*where* $r$ *is the radius of* $\mathcal{G}$, *meaning that tokens converge to a common vector exponentially.*

The detailed proof is provided in Appendix B. The above result suggests that with pure self-attention, as long as there is a token which all other tokens in the sequence can directly or indirectly attend to over a fixed number of layers, exponential rank collapse of tokens to a common vector is guaranteed. In particular, it generalizes the main result in [12] from $\mathcal{G}$ being a complete graph to a much more general class of attention patterns: the attention pattern $\mathcal{G}$ only needs to be quasi-strongly connected, meaning that the result applies to general attention masks used in practice including the causal mask used in decoder-only models such as the GPT family [6, 30], or sparse attention patterns deployed in many efficient transformer models [4, 10, 21, 32, 42]. We discuss a few interesting implications below.

**Local vs. global attention**    The exponential rate $(1 - \epsilon^r)^{1/r}$ is monotone in the graph radius $r$. This means that rank collapse should be slower for graphs with larger radius $r$. Our result thus indirectly supports the use of local attention patterns [4, 42], which not only make the attention computation more efficient (what those works are originally motivated by), but also implicitly alleviate the rank collapse issue.

**Focused vs. uniform attention**    In addition, the exponential rate is monotone decreasing in $\epsilon$, which means that rank collapse is slower with smaller $\epsilon$. One can interpret $\epsilon$ as how "focused" attention is distributed among reachable tokens, as $\epsilon$ is maximized when attention happens uniformly among reachable tokens. Besides applying attention masks and restricting the number of reachable tokens, another way to control how focused attention would be is through the temperature term $d_{QK}$. As larger values of $d_{QK}$ would make the attention allocation among reachable tokens more even, they should make rank collapse happen faster across layers.

**Trade-off between rank collapse and universal approximating power**    Finally, for strongly connected graphs, the above result also reveals a trade-off between universal function approximation power and the rate of rank-collapse. Yun et al. [41] show that transformers with strongly connected graph masks are sequence-to-sequence function universal, yet with a mask $\mathcal{G}$ they need at least the diameter of $\mathcal{G}$ layers to achieve the full sequence-to-sequence function approximation property. This implies that masks with smaller diameters (and thus smaller radii, as radius $\leq$ diameter $\leq 2$ radius) are more efficient in terms of function approximation power, yet they are more prone to rank collapse.

**Remark 1.** *Since the analysis in [12] fundamentally relies on the shift-invariant property of* $\text{softmax}(\cdot)$, *it is necessary that all the tokens are allowed to attend to all the other tokens for their proof to work. On the contrary, we leverage a different graph-theoretic approach to exploit the common structure of the attention matrices to obtain a general result for masked attention.*

## 4.2 Masked Attention with LayerNorm: Rank Collapse

So far, we have considered the pure self-attention dynamics without LayerNorm and focused on the role of the attention mask. What happens if we add LayerNorm and consider the attention dynamics defined in (3) instead? In this section, we first present a negative result, showing that exponential collapse of tokens to a common vector can still happen for certain classes of value matrices.

**Theorem 2.** *Consider the self-attention dynamics with LayerNorm defined in* (3). *Let $\mathcal{G}$ be a strongly connected graph. Assume* **A1**-**A2***, and that $W_V^{(t)}$ is orthogonal for all $t \geq 0$, and in addition, the initial input $X^{(0)}$ satisfies that*

$(*)$ $N \leq d$, $X^{(0)}$ *has full rank.*

*Then there exist $C > 0$, $\epsilon > 0$ such that $N\epsilon < 1$ and*

$$\mu(X^{(t)}) \leq C(1 - N\epsilon^{2r})^{\frac{t}{2r}} \qquad \forall t \geq 0 \,, \tag{7}$$

*where $r$ is the radius of $\mathcal{G}$, meaning that tokens converge to a common point on $\mathbb{S}^{d-1}$ exponentially.*

The detailed proof is provided in Appendix C. The result can be seen as a generalized discrete version of Theorem 4.1 in [15]. Notably, our analysis is based purely on advanced linear algebra tools: infinite products of non-negative matrices and their ergodicity, and can account for time-varying weights and general attention masks, as opposed to fixed $W_K, W_Q, W_V$ over time and $\mathcal{G}$ being complete (which is the case in [15]). One way to satisfy the condition $(*)$ on the initial input $X^{(0)}$ is to require $N \leq d$ and to initialize tokens uniformly randomly on $\mathbb{S}^{d-1}$, then the condition hold almost surely. This is how the condition is dealt with in [15].

Note that the condition $(*)$ implies that there exists $v \in \mathbb{S}^{d-1}$ such that $\langle X_{i,:}^{(0)}, v \rangle > 0$ for all $i \in [N]$ by either the hyperplane separation theorem or Farkas' lemma (see Lemma 6 in Appendix C). If the initial token geometry satisfies a stronger condition than the above, then $(*)$ is no longer necessary and Theorem 2 even directly generalizes to quasi-strongly connected graphs $\mathcal{G}$. We define $\phi^{(t)} := \min_{i,j \in [N]} \langle X_{i,:}^{(t)}, X_{j,:}^{(t)} \rangle$, indicating the minimal cosine similarity between tokens. If the cosine similarities are non-negative for all pairs of tokens initially, then the rank collapse happens exponentially, as long as $\mathcal{G}$ is quasi-strongly connected.

**Corollary 1.** *Consider the self-attention dynamics with LayerNorm defined in* (3). *Let $\mathcal{G}$ be a quasi-strongly connected graph. Under* **A1**-**A2***, if $W_V^{(t)}$ is orthogonal for all $t \geq 0$ and $\phi^{(0)} \geq 0$, then there exist $C > 0$, $\epsilon > 0$ such that $N\epsilon < 1$ and*

$$\mu(X^{(t)}) \leq C(1 - \epsilon^{2r})^{\frac{t}{2r}}, \qquad \forall t \geq 0 \,. \tag{8}$$

*where $r$ is the radius of $\mathcal{G}$, meaning that tokens converge to a common point on $\mathbb{S}^{d-1}$ exponentially.*

**Full mask vs. causal mask**    We can refine Corollary 1 by specifying the number of center nodes $n$ in the mask $\mathcal{G}$, then the upper bound for the exponential rate would be $(1 - n\epsilon^{2r})$ instead, meaning that the rate of rank collapse can be negatively affected by the number of center nodes in the mask $\mathcal{G}$. In the case of full attention where $\mathcal{G}$ is the complete graph, the mask would have $N$ center nodes, matching the bound in Theorem 2. In the case of causal attention where $\mathcal{G}$ is the causal graph, the mask would only have one center node, and the upper bound would be looser, suggesting the advantage of the causal mask in mitigating the rate of rank collapse as compared to the full mask.

**Post-LN vs. Pre-LN**    The definition of LayerNorm in (3) follows from the original transformer paper [35] and nowadays it is referred as *post-LN* [39]. An alternative use of LayerNorm in many LLMs, where LayerNorm comes before self-attention, is called *pre-LN* and can be written instead as

$$X^{(t+1)} = A^{(t)} \operatorname{LN}(X^{(t)}) W^{(t)} := A^{(t)} D^{(t)} X^{(t)} W_V^{(t)} \,. \tag{9}$$

Note that Theorem 2 and Corollary 1 apply directly to the case of pre-LN with similar proofs.

### 4.3   Masked Attention with LayerNorm: Counterexample

The main results from the previous sections seem pessimistic: the self-attention dynamics seem doomed to collapse into a rank one subspace in the long run, with or without LayerNorm. In this section, however, we will first construct a nontrivial counterexample with only LayerNorm such that for a general class of input sequences, tokens converge to an equilibrium where rank collapse does not happen. Notice that for a transformer model to be practical, it is important that it can prevent rank collapse for a general class of input sequences rather than a specific input sequence. We will then show a general result stating that, with LayerNorm and proper choice of value matrices, the self-attention dynamics can have a rich set of equilibria with *any possible rank* between one and full. Moreover, for a general class of input sequences, tokens provably do not converge to a rank one subspace under the resultant dynamics.

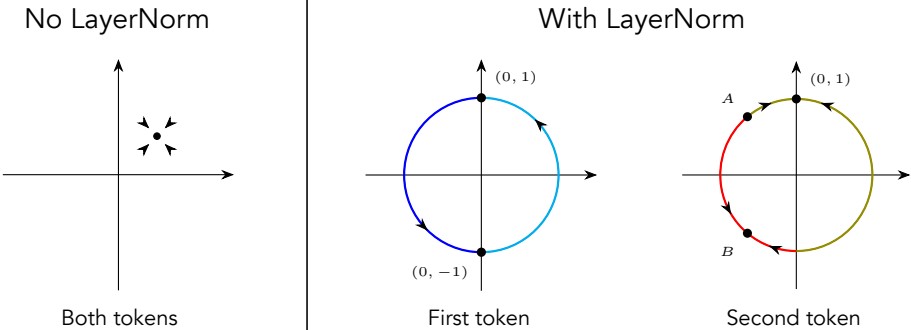

Figure 1: Long-term behavior of tokens in the case of $N = 2, d = 2$. Without LayerNorm (left), both tokens collapse to the same point in $\mathbb{R}^2$; whereas with LayerNorm (right), such a collapse would not necessarily happen and token representations can maintain full rank in the long term (first token converges either to $(0, 1)$ or $(0, -1)$. Assuming convergence to $(0, 1)$ for the first token, the second token converges to $B$, if it is initially located within the red segment).

### 4.3.1 Illustrative Counterexample

For simplicity of illustration, we consider $N = 2, d = 2$, and $\mathcal{G}$ to be the causal mask. Then let $W_K^{(t)} = W_Q^{(t)} = \mathbf{0}$, which leads to the attention matrices

$$A^{(t)} = \begin{bmatrix} 1 & 0 \\ 1/2 & 1/2 \end{bmatrix} \quad \forall t \geq 0.$$

We further let

$$W_V^{(t)} = \begin{bmatrix} 1 & w \\ 0 & 1 \end{bmatrix} \quad \forall t \geq 0,$$

for $w \in \mathbb{R}$. Without loss of generality, fix $w > 1$. Then a careful analysis shows that, depending on its initial position, the first token will either converge to $(0, 1)$ or $(0, -1)$. Suppose the first token converges to $(0, 1)$. Then the convergence of the second token as $t \to \infty$ is illustrated in Fig. 1, where $A = \left(-\frac{1}{w}, \sqrt{1 - \frac{1}{w^2}}\right)$, $B = \left(-\frac{1}{w}, -\sqrt{1 - \frac{1}{w^2}}\right)$. A rigorous proof can be found in Appendix E. Note that due to the scaling effect of LayerNorm, any scaled version $c^{(t)} W_V^{(t)}, c^{(t)} > 0$ of $W_V^{(t)}$ works equivalently here.

**Remark 2.** *For any orthogonal $Z \in \mathbb{R}^{d \times d}$, $W_Z = Z^\top W Z$ works equivalently in this example, and the resulting token trajectories follow $X^{(t)} Z$.*

This nontrivial counterexample suggests that with the LayerNorm dynamics in (3), there are proper choices for $W_K^{(t)}, W_Q^{(t)}, W_V^{(t)}$ matrices that can prevent tokens from collapsing to a rank one subspace, for a nonzero measure set of input sequences.

### 4.3.2 Generalization of the Counterexample

We conclude this section with the following statement generalizing the previous illustrative example: with a proper choice of weight matrices, tokens under the attention dynamics with LayerNorm can simultaneously have equilibria of any rank between one and full. Moreover, for a general class of input sequences, tokens provably do not converge to a rank one subspace.

**Theorem 3.** *Let $\mathcal{G}$ be the causal graph and consider the attention dynamics with LayerNorm defined in (3). Then there exists $\left\{W_Q^{(t)}, W_K^{(t)}, W_V^{(t)}\right\}_{t=0}^{\infty}$ satisfying **A2**-**A3** such that the corresponding dynamic has the following properties:*

1. *For any $1 \leq k \leq \min\{N, d\}$, there exist at least $2^k$ equilibria of rank $k$;*

2. *For a general class of input sequences $X^{(0)}$ in $(\mathbb{S}^{d-1})^N$ with measure greater than $0$, $X^{(t)}$ does not converge to a rank one subspace as $t \to \infty$.*

The detailed proof is provided in Appendix F. For the sake of simplicity and observing that in a causal graph, there is an inherent chronological order among tokens, in the convergence analysis of each token we assume all tokens preceding it have already converged.

The above result is in direct contrast to the case without LayerNorm (Theorem 1), where all input sequences eventually result in complete rank collapse to a one-dimensional subspace. It also stands in contrast to the hypothesis that LayerNorm plays no role in mitigating the collapse [12], and suggests that it is an important component of the self-attention dynamics in transformers—adding LayerNorm fundamentally changes the behavior of the underlying system. Compared to the case without LayerNorm, first, LayerNorm guarantees that the system never diverges, no matter what the value matrices are. Second, and more importantly, attention with LayerNorm leads to a surprisingly more expressive, versatile dynamics than the system without it—composition with different value matrices $W_V^{(t)}$ can result in different long-term token behaviors: in some cases, tokens completely collapse to a single point (Theorem 2), while in others tokens can retain higher rank asymptotically (Theorem 3).

### 4.4 Discussion

Next, we discuss three further intriguing findings from our analysis of the self-attention dynamics with LayerNorm presented in the previous section.

**Scaling effect of LayerNorm is key**  From a dynamical system perspective, rank collapse happens under pure self-attention because the attraction of the center node causes all the tokens to be aligned. In the counterexample provided in Section 4.3.1, the key insight to prevent the second token from aligning with the first token is that the updated representation of the second token must generate a component canceling out the attraction from the first token, which acts as a repulsion. The crucial role of LayerNorm here is to stabilize this cancellation process by readjusting the scale of tokens to ensure that the cancellation persists in all the updates, which would not be the case if there is no LayerNorm.

**Anisotropy of token embeddings**  Analyzing and understanding the token geometry in transformers has long been of interest to the NLP and general ML community. In particular, empirical findings suggest that contextual token embeddings generated by transformers are anisotropic, meaning they tend to concentrate in a narrow region [1, 9, 13, 14, 17]. Interestingly, as we show in Appendix F.2, the full rank equilibria described in Theorem 3 align with this observation, as tokens lie in a narrow region on $\mathbb{S}^{d-1}$. While it still remains unclear how exactly such a representation geometry is useful for downstream tasks [1, 9, 14, 17], empirical studies have found that anisotropy is not necessarily harmful for semantic representations and can assist with tasks like clustering [1, 26].

**Stability of the equilibria**  Finally, we would like to note that due to the combinatorial nature of the analysis with increasing dimensions, we do not prove the exact convergence to specific equilibria beyond not converging to a rank one subspace. However, we observe in simulation that these equilibria are indeed stable in certain directions, despite their regions of attraction being relatively small. This suggests that while LayerNorm improves the expressivity of pure self-attention dynamics, the resulting expressive power might still be limited in a way that it would be difficult for a generic input sequence to reach a specific configuration at equilibrium. This observation seems in line with the fact that MLP modules are crucial for the universal sequence-to-sequence approximation power of transformers [40, 41]. To achieve maximal expressivity, and in particular, to be able to move tokens freely around the whole space (such as in the case for transformers with MLPs), nonlinear power from MLPs would be necessary.

## 5  Numerical Experiments

In this section, we validate our theoretical findings via numerical experiments. Following [12], we randomly select 3000 samples of 128-token excerpts (based on the BERT tokenizer) from Wikipedia using the Wikipedia API in Python. More experimental details and additional results can be found in Appendix G.

**The effect of attention masks and LayerNorm** We first verify the effect of different attention masks and LayerNorm on rank collapse with respect to $\mu(\cdot)$. We use BERT [11] as the backbone transformer model and consider five different model variants for a controlled experiment: self-attention network (SAN) in which the model has self-attention layers; SAN with skip connections; SAN with LayerNorm where in each layer there is self-attention followed by LayerNorm; SAN with both skip connections and LayerNorm where LayerNorm is put after skip connections; and finally the standard transformer with all the components. For attention masks, we select four representative types: the complete graph, the causal graph, the sliding window (tokens can only attend to the token right before and after and themselves) and the uni-directional sliding window (tokens can only attend to the token right before and themselves). Fig. 2 shows the average values of $\mu(X^{(t)})$ with standard deviations over the 3000 samples in different architectures. We see that in SANs, $\mu(X^{(t)})$ converges to zero exponentially for all attention masks, yet more local attention masks slow down the convergence rate. Furthermore, in randomly initialized models, right after we add in LayerNorm, $\mu(X^{(t)})$ no longer converges to zero and can be maintained at a stable level significantly above zero. In pretrained models, LayerNorm helps prevent the issue together with other components such as skip connections and stabilize the representations. Results in 128-layer initialized models also exhibit the same trend, and can be found in Appendix G.1.

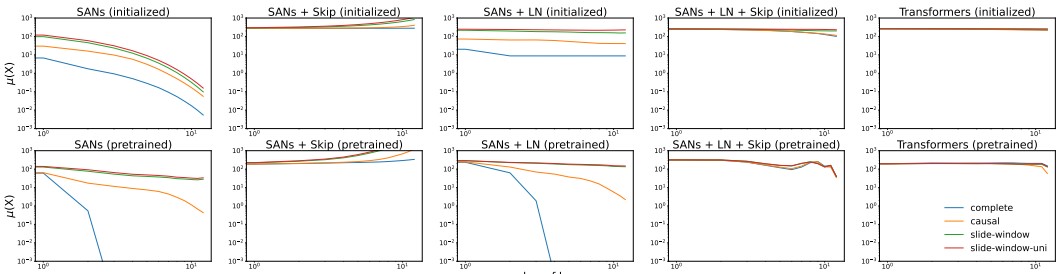

Figure 2: Evolution of $\mu(X^{(t)})$ (in log-log scale) as the number of layers increases. Rank collapse happens exponentially for pure attention, despite different attention masks having different convergence rates. However, as soon as we solely add in LayerNorm, $\mu(X^{(t)})$ no longer converge to zero in randomly initialized models; in pretrained models, LayerNorm helps prevent the issue together with other components and stabilize the representations.

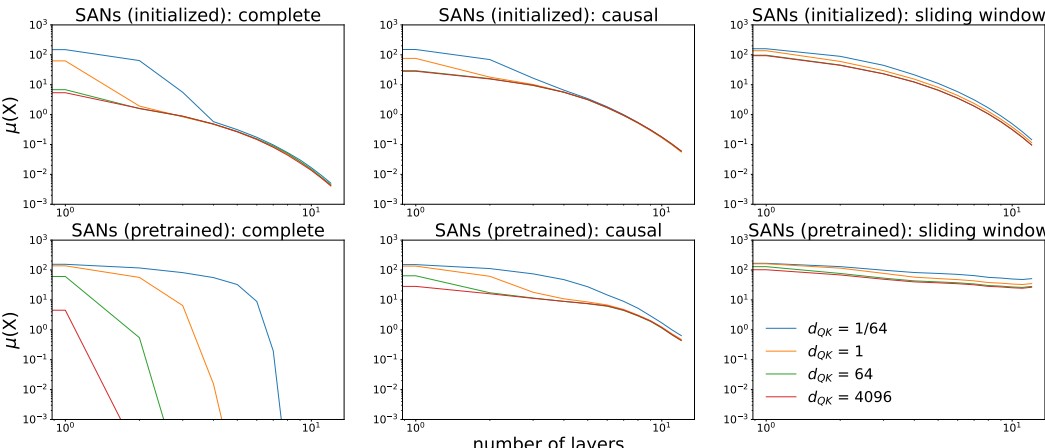

Figure 3: Evolution of $\mu(X^{(t)})$ (in log-log scale) as the number of layers increases. Smaller temperature terms alleviate the rate of rank collapse, and effect is more significant with global attention than with sparser masked attention, and more in shallower layers than deeper layers.

**The complex interplay between different components in transformers** Previous works have shown that skip connections can help combat rank collapse in transformers. Yet in Fig. 2, skip connections seem to make $\mu(X)$ unstable, particularly in deeper layers (for 128-layer results, see Appendix G.1). Compared with full transformers where $\mu(X)$ stays relatively stable, there is a clear

discrepancy. In that case, LayerNorm emerges as a crucial element in mitigating rank collapse and stabilizing token representations while also counteracting potential negative effects of pure skip connections. This underscores the complex interplay between different components in transformers.

**The effect of the temperature term**     Next, we investigate the effect of the temperature term $d_{QK}$ in the attention calculation. Since the amount of focus of the attention is affected by both the attention mask and the temperature term, we investigate the effect of $d_{QK}$ with different attention masks. Fig. 3 shows the average values of $\mu(X^{(t)})$ with standard deviations over the 3000 samples with different masks in pretrained SANs. We observe that while a smaller temperature term $d_{QK}$ alleviates the (initial) rate of rank collapse in all cases, the effect is more significant with global attention than with sparser masked attention, and more in shallower layers than in deeper layers. The results in initialized models show similar trends and can be found in Appendix G.2.

**Evolution of token geometry**     Finally, we study the evolution of token geometry as the number of layers increases in transformers. Specifically, to capture more fine-grained details than $\mu(\cdot)$, we measure how the rank, minimal singular value of token representations, and absolute values of pairwise cosine similarities among tokens change as tokens pass through the transformer layers. We select four representative pretrained transformer models: BERT [11], GPT2 [30], T5 [31] and ALBERT [24] and the average results over 3000 samples are shown in Fig. 4. We see that all models exhibit similar behaviors as tokens evolve along the layers: the models can effectively preserve the full rank of token representations, as also evident from the minimal singular values consistently remaining significantly above zero. Yet the absolute cosine similarities among tokens suggest that tokens gradually concentrate in a narrow region. This empirical observation aligns with the characteristics of the equilibrium in our counterexample that a stable long-term geometry for token representations in transformers can retain full rank while being close to a low dimensional geometry at the same time.

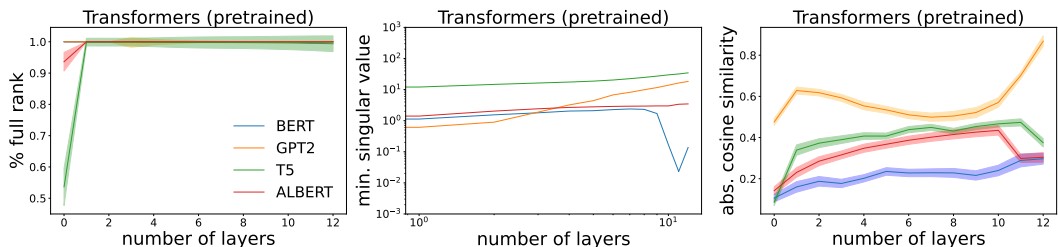

Figure 4: Evolution of token geometry as the number of layers increases. We see that tokens are indeed able to maintain full rank, while at the same time the representations are anisotropic, meaning that they concentrate in a narrow region, as indicated by the average pairwise absolute cosine similarities.

## 6   Conclusion

The attention mechanism has led to significant empirical progress in building foundation models. From a dynamical system perspective, the time-varying and nonlinear nature of self-attention dynamics, when coupled with attention masks, LayerNorm and other components in transformers, enables expressive and complex behavior. In particular, we show that the choice of attention masks directly affects the token dynamics. It would hence be valuable for future research to investigate how to effectively design attention masks. Our result further suggests that robust token geometries under multi-layer self-attention can exhibit both full-rankness and anisotropic characteristics simultaneously, which is the case in real transformers as well. It would be interesting to study how such co-existence of these two characteristics would help with learning tasks — whether the full-rank yet close-to-low-dimensional geometry enables efficient learning and generalization while still letting tokens capture meaningful fine-grained details.

## Acknowledgments

X.W., A.A., and A.J. would like to thank Bernard Chazelle for helpful discussions. X.W., A.A., and A.J. were supported by ONR Award N00014-23-1-2299 and a Vannevar Bush fellowship from the

Office of the Under Secretary of Defense for Research and Engineering (USD(R&E)). Y.W. and S.J. were supported by ONR Award N00014-20-1-2023 (MURI ML-SCOPE), NSF AI Institute TILOS (NSF CCF-2112665), NSF Award 2134108, and the Alexander von Humboldt Foundation.

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

## A    Comment about Section 3.3 in Dong et al. [12]

From the mathematical form of LayerNorm in Eq. (3), since LayerNorm normalizes each row of $\tilde{X}^{(t)}$ to have 2-norm one, the operator that represents this scaling effect $D^{(t)}$ should be multiplied from the left hand side (scaling each row) rather than from the right hand side (scaling each column). In particular, since in Dong et al. [12], token representation $X$ takes the same formulation that rows represent different tokens while columns represent different features, the same rule applies. Thus, the argument based on $D^{(t)}$ being multiplied from the right hand side and can be merged with $W_V^{(t)}$ would not work.

## B    Proof of Theorem 1

### B.1    Auxiliary results

**Lemma 1.** *Under* **A1**-**A3***, there exists $\epsilon > 0$ such that $A_{i,j}^{(t)} \geq \epsilon$ for all $t \geq 0$, $(j,i) \in E$.*

*Proof.* Writing (2) recursively, we get that the token trajectories

$$X^{(t+1)} = A^{(t)}...A^{(0)}X^{(0)}W_V^{(0)}...W_V^{(t)} , \tag{10}$$

stay uniformly bounded for all $t \geq 0$ by **A3**. Then it follows from **A2** that there exists $C \in \mathbb{R}$ such that for all $t \geq 0$,

$$\left\| \left( X^{(t)}W_Q^{(t)} \right)_{i,:} \right\|_2 = \left\| X_{i,:}^{(t)}W_Q^{(t)} \right\|_2 \leq C ,$$

$$\left\| \left( X^{(t)}W_K^{(t)} \right)_{i,:} \right\|_2 = \left\| X_{i,:}^{(t)}W_K^{(t)} \right\|_2 \leq C .$$

Hence for all $i, j \in [N]$,

$$-C^2 \leq (X^{(t)}W_Q^{(t)}(X^{(t)}W_K^{(t)})^\top)_{i,j} \leq C^2 .$$

This implies that there exists $\epsilon > 0$ such that $A_{i,j}^{(t)} \geq \epsilon$ for all $(j,i) \in E$.                                                                               □

Fix a vector $x^{(0)} \in \mathbb{R}^d$ and a $\{A^{(n)}\}_{n=0}^{\infty}$ in $\mathcal{A}_{\mathcal{G}, \epsilon}$. Let $x^{(t)} := A^{(t)} \dots A^{(0)} x^{(0)}$. We further denote that

$$\max_{i \in d} x_i^{(t)} := M^{(t)}, \qquad \min_{i \in d} x_i^{(t)} := m^{(t)}.$$

Note that $M^{(t)}$ is monotone non-increasing in $t$, whereas $m^{(t)}$ is monotone non-decreasing in $t$.

**Lemma 2.** *Let* $x_i^{(t)} = pm^{(t)} + (1-p)M^{(t)}$ *for* $0 \le p \le 1$. *Then for any* $T \in \mathbb{N}$,

$$x_i^{(t+T)} \le p \left( \prod_{n=t}^{t+T-1} A_{ii}^{(n)} \right) m^{(t)} + \left( 1 - p \left( \prod_{n=t}^{t+T-1} A_{ii}^{(n)} \right) \right) M^{(t)}, \tag{11}$$

*and*

$$x_i^{(t+T)} \ge p \left( \prod_{n=t}^{t+T-1} A_{ii}^{(n)} \right) M^{(t)} + \left( 1 - p \left( \prod_{n=t}^{t+T-1} A_{ii}^{(n)} \right) \right) m^{(t)}. \tag{12}$$

*Proof.* Given $x_i^{(t)} = pm^{(t)} + (1-p)M^{(t)}$, we get that

$$\begin{aligned}
x_i^{(t+1)} &= \sum_{j=1}^{N} A_{ij}^{(t)} x_j^{(t)} \\
&\le A_{ii}^{(t)}(pm^{(t)} + (1-p)M^{(t)}) + (1 - A_{ii}^{(t)})M^{(t)} \\
&= pA_{ii}^{(t)} m^{(t)} + (1 - pA_{ii}^{(t)})M^{(t)}.
\end{aligned}$$

Subsequently,

$$\begin{aligned}
x_i^{(t+2)} &= \sum_{j=1}^{N} A_{ij}^{(t+1)} x_j^{(t+1)} \\
&\le A_{ii}^{(t+1)}(pA_{ii}^{(t)} m^{(t)} + (1 - pA_{ii}^{(t)})M^{(t)}) + (1 - A_{ii}^{(t+1)})M^{(t)} \\
&= p \left( \prod_{n=t}^{t+1} A_{ii}^{(s)} \right) m^{(t)} + \left( 1 - p \left( \prod_{n=t}^{t+1} A_{ii}^{(s)} \right) \right) M^{(t)}.
\end{aligned}$$

We obtain (11) by iterating the process. Similarly, (12) can be derived using a symmetric argument as for the upper bound. $\square$

**Lemma 3.** *Let* $\{A^{(n)}\}_{n=0}^{\infty}$ *in* $\mathcal{A}_{\mathcal{G}, \epsilon}$ *and* $r$ *be the radius of* $\mathcal{G}$. *Then for* $t \ge 0$ *and* $0 \le p \le 1$,

$$M^{(t+kr)} - m^{(t+kr)} \le (1 - p\epsilon^r)^k \left( M^{(t)} - m^{(t)} \right), \quad k \in \mathbb{N}_{\ge 0}. \tag{13}$$

*Proof.* Let $i_0 \in \mathcal{G}$ be a center token and fix $t \ge 0$ and $0 \le p \le 1$. Without loss of generality, suppose

$$x_{i_0}^{(t)} \le pm^{(t)} + (1-p)M^{(t)}. \tag{14}$$

From Lemma 2, we get that for all $T \in \mathbb{N}$,

$$\begin{aligned}
x_{i_0}^{(t+T)} &\le p \left( \prod_{s=t}^{t+T-1} A_{i_0 i_0}^{(s)} \right) m^{(t)} + \left( 1 - p \left( \prod_{s=t}^{t+T-1} A_{i_0 i_0}^{(s)} \right) \right) M^{(t)} \\
&\le p\epsilon^T m^{(t)} + \left( 1 - p\epsilon^T \right) M^{(t)}.
\end{aligned}$$

Denote $\mathcal{V}_k$ as the set of tokens that are exactly $k$-hop neighbors of $i_0$:

$$\mathcal{V}_k := \{ j \in \mathcal{V} : \mathrm{dist}(i_0, j) = k \}.$$

For any $i_1 \in \mathcal{V}_1$, it follows that

$$\begin{aligned}
x_{i_1}^{(t+1)} &\le A_{i_1 i_0}^{(t)} x_{i_0}^{(t)} + (1 - A_{i_1 i_0}^{(t)})M^{(t)} \\
&\le \epsilon \left( pm^{(t)} + (1-p)M^{(t)} \right) + (1 - \epsilon)M^{(t)} \\
&= p\epsilon m^{(t)} + (1 - p\epsilon) M^{(t)}.
\end{aligned} \tag{15}$$

Apply Lemma 2 to (15), we further get that for $T \in \mathbb{N}$,

$$x_{i_1}^{(t+T)} \leq p\epsilon^T m^{(t)} + \left(1 - p\epsilon^T\right) M^{(t)} .$$

Then let $i_2 \in \mathcal{V}_2$. For simplicity, we still use $i_1$ to denote the intermediate neighbor between $i_0$ and $i_2$. Similarly to the case of $i_1$, we get that

$$\begin{aligned}
x_{i_2}^{(t+2)} &\leq A_{i_2 i_1}^{(t+1)} x_{i_1}^{(t+1)} + (1 - A_{i_2 i_1}^{(t+1)}) M^{(t)} \\
&= \epsilon \left( p\epsilon m^{(t)} + (1 - p\epsilon) M^{(t)} \right) + (1 - \epsilon) M^{(t)} \\
&= p\epsilon^2 m^{(t)} + \left(1 - p\epsilon^2\right) M^{(t)} ,
\end{aligned} \tag{16}$$

and applying Lemma 2 to (16), it follows that for $T \in \mathbb{N}_{\geq 2}$

$$x_{i_2}^{(t+T)} \leq p\epsilon^T m^{(t)} + \left(1 - p\epsilon^T\right) M^{(t)} .$$

Iterating this process for tokens in $\mathcal{V}_3, ..., \mathcal{V}_r$, where $r$ is the radius of $\mathcal{G}$, we get that for any token $i \in \mathcal{G}$,

$$x_i^{(t+r)} \leq p\epsilon^r m^{(t)} + \left(1 - p\epsilon^r\right) M^{(t)} ,$$

meaning that

$$M^{(t+r)} \leq p\epsilon^r m^{(t)} + \left(1 - p\epsilon^r\right) M^{(t)} .$$

We thus conclude that

$$\begin{aligned}
M^{(t+r)} - m^{(t+r)} &\leq p\epsilon^r m^{(t)} + (1 - p\epsilon^r) M^{(t)} - m^{(t)} \\
&= (1 - p\epsilon^r) \left( M^{(t)} - m^{(t)} \right)
\end{aligned} \tag{17}$$

For the other case of (14), i.e. $x_{i_0}^{(t)} > pm^{(t)} + (1 - p)M^{(t)}$, (17) is obtained using a symmetric argument by bounding $m^{(t+r)}$ from below through (12). This completes the proof of (13). $\qquad\square$

**Corollary 2.** *There exists $C(x^{(0)}) > 0$ such that*

$$M^{(t)} - m^{(t)} \leq C \left(1 - \epsilon^r\right)^{t/r} , \forall t \geq 0 ,$$

*where $r$ is the radius of the graph $\mathcal{G}$.*

### B.2 Proof of Theorem 1

With Corollary 2, we are ready to prove the main theorem. Recall (10), we then get that

$$\begin{aligned}
X_{:,j}^{(t+1)} &= (A^{(t)} ... A^{(0)} X^{(0)} W_V^{(0)} ... W_V^{(t)})_{:,j} \\
&= A^{(t)} ... A^{(0)} X^{(0)} \left( W_V^{(0)} ... W_V^{(t)} \right)_{:,j} \\
&= \sum_{i=1}^d \tilde{W}_{ij}^{(t)} A^{(t)} ... A^{(0)} X_{:,i}^{(0)}
\end{aligned} \tag{18}$$

where $\tilde{W}^{(t)} := W_V^{(0)} ... W_V^{(t)}$. This means that for any $j \in [d]$, there exists $C_j > 0$ such that

$$\begin{aligned}
\left| X_{m,j}^{(t+1)} - X_{n,j}^{(t+1)} \right| &= \left| \sum_{i=1}^d \tilde{W}_{ij} ((A^{(t)} ... A^{(0)} X_{:,i}^{(0)})_m - (A^{(t)} ... A^{(0)} X_{:,i}^{(0)})_n) \right| \\
&\leq \sum_{i=1}^d \left| \tilde{W}_{ij} \right| \left| (A^{(t)} ... A^{(0)} X_{:,i}^{(0)})_m - (A^{(t)} ... A^{(0)} X_{:,i}^{(0)})_n \right| \\
&\leq \sum_{i=1}^d \left| \tilde{W}_{ij} \right| \left| \max_{l \in [N]} (A^{(t)} ... A^{(0)} X_{:,i}^{(0)})_l - \min_{l \in [N]} (A^{(t)} ... A^{(0)} X_{:,i}^{(0)})_l \right| \\
&\leq C_j \sum_{i=1}^d \left| \tilde{W}_{ij} \right| (1 - \epsilon^r)^{t/r} \tag{19} \\
&\leq C_j (1 - \epsilon^r)^{t/r} \quad \forall m, n \in [N] , \tag{20}
\end{aligned}$$

where (19) follows from (2) and (20) follows from **A3**.

Then by (20),

$$\mu(X^{(t)}) = \|X^{(t)} - \mathbf{1}\mathbf{1}^\top X^{(t)}/N\|_F = \sqrt{\sum_{j=1}^{d} \|X_{:,j}^{(t+1)} - \mathbf{1}\mathbf{1}^\top X_{:,j}^{(t)}/N\|_2^2}$$

$$= \sqrt{\frac{1}{2N}\sum_{j=1}^{d}\sum_{m=1}^{N}\sum_{n=1}^{N}|X_{m,j}^{(t)} - X_{n,j}^{(t)}|^2}$$

$$\leq \sqrt{\frac{N}{2}\sum_{j=1}^{d}C_j^2\left(1-\epsilon^r\right)^{2t/r}} \leq \sqrt{C\left(1-\epsilon^r\right)^{2t/r}}$$

$$= C'\left(1-\epsilon^r\right)^{t/r},$$

where $C := \frac{N}{2}\sum_{j=1}^{d}C_j^2$ and $C' := \sqrt{C}$ here.

## C  Proof of Theorem 2

### C.1  Auxiliary results

**Lemma 4.** *Under **A1** and **A2**, there exists $\epsilon > 0$ such that $A_{i,j}^{(t)} \geq \epsilon$ for all $t \geq 0$, $(j,i) \in E$.*

*Proof.* Note due to the way the dynamical system is defined in (3), for every $t \geq 0$, every token $X_{i,:}^{(t)}$ lies within the unit sphere $\mathbb{S}^{d-1}$. Then it follows from **A2** that there exists $C \in \mathbb{R}$ such that for all $t \geq 0$,

$$\left\|\left(X^{(t)}W_Q^{(t)}\right)_{i,:}\right\|_2 = \left\|X_{i,:}^{(t)}W_Q^{(t)}\right\|_2 \leq C,$$

$$\left\|\left(X^{(t)}W_K^{(t)}\right)_{i,:}\right\|_2 = \left\|X_{i,:}^{(t)}W_K^{(t)}\right\|_2 \leq C.$$

Hence for all $i,j \in [N]$,

$$-C^2 \leq (X^{(t)}W_Q^{(t)}(X^{(t)}W_K^{(t)})^\top)_{i,j} \leq C^2.$$

This implies that there exists $\epsilon > 0$ such that $A_{i,j}^{(t)} \geq \epsilon$ for all $(i,j) \in E$. $\qquad\square$

**Lemma 5.** *Under the same assumptions as Theorem 2, there exists $C_1, C_2 > 0$ such that $C_1 \leq D_{i,i}^{(t)} \leq C_2$ for all $t \geq 0$, $i \in [N]$.*

*Proof.* We adopt the following notation:

**Notation 1.** *For $X \in \mathbb{R}^{N\times d}$, we use $\mathrm{Conv}(X)$ to denote the convex hull of $\{X_{1,:}, ..., X_{N,:}\}$.*

First, since for $x \in \mathrm{Conv}(X^{(t)})$, $W^{(t)\top}x$ always lies in $\mathbb{S}^{d-1}$, $D_{i,i}^{(t)} \geq 1$.

Then for all $t \geq 0$, each row vector of $(A^{(t)}X^{(t)}W^{(t)})_{i,:}$ lies in the convex cone generated by $X^{(0)}\tilde{W}^{(t)}$, where $\tilde{W}^{(t)} := \Pi_{i=0}^{t}W^{(i)}$. To see this,

$$(A^{(t)}X^{(t)}W^{(t)})_{i,:} = \sum_{j_0=1}^{N}A_{ij_0}^{(t)}X_{j_0,:}^{(t)}W^{(t)}$$

$$= \sum_{j_0=1}^{N}A_{ij_0}^{(t)}D_{j_0,j_0}^{(t-1)}\left(\sum_{j_1=1}^{N}A_{j_0,j_1}^{(t-1)}X_{j_1,:}^{(t-1)}W^{(t-1)}\right)W^{(t)}$$

$$= \sum_{(j_0,...,j_t)\in[N]^{t+1}}A_{ij_0}^{(t)}D_{j_0,j_0}^{(t-1)}A_{j_0,j_1}^{(t-1)}D_{j_1,j_1}^{(t-2)}...D_{j_{t-1},j_{t-1}}^{(0)}A_{j_{t-1},j_t}^{(0)}X_{j_t,:}^{(0)}\tilde{W}^{(t)}$$

Hence all $t \geq 0$,

$$(A^{(t)}X^{(t)}W^{(t)})_{i,:} = \sum_{j=1}^{N} a_{ij}^{(t)} X_{j,:}^{(0)} \tilde{W}^{(t)},$$

where $a_{ij}^{(t)} \geq 0, \forall j \in [N]$ and $\sum_{j=1}^{N} a_{ij}^{(t)} \geq 1$.

We then make the following observation:

**Lemma 6.** *Under assumption (∗), there exists $v \neq \mathbf{0}$ such that*

$$\langle X_{i,:}^{(0)}, v \rangle > 0 \quad i \in [N],$$

*meaning that all the points of $X^{(0)}$ lie in an open hemisphere $\mathcal{H}$.*

To see this, observe that since $X^{(0)}$ is full rank and $N \leq d$, $\mathbf{0} \notin \text{Conv}(X^{(0)})$. Then the hyperplane separation theorem states that there exists a hyperplane strictly separating $\mathbf{0}$ and $\text{Conv}(X^{(0)})$, letting us conclude Lemma 6.

Since all the points of $X^{(0)}$ lie in an open hemisphere $\mathcal{H}$, there exists $\gamma > 0$ such that for all $x \in \text{Conv}(X^{(0)})$ and $W$ orthogonal,

$$\|Wx\|_2 = \|x\|_2 \geq \gamma.$$

Then

$$
\begin{aligned}
\|(A^{(t)}X^{(t)}W^{(t)})_{i,:}\|_2 &= \left\| \left( \sum_{j=1}^{N} a_{ij}^{(t)} \right) \frac{\sum_{j=1}^{N} a_{ij}^{(t)} X_{j,:}^{(0)} \tilde{W}^{(t)}}{\sum_{j=1}^{N} a_{ij}^{(t)}} \right\|_2 \\
&\geq \left\| \frac{\sum_{j=1}^{N} a_{ij}^{(t)} X_{j,:}^{(0)} \tilde{W}^{(t)}}{\sum_{j=1}^{N} a_{ij}^{(t)}} \right\|_2 \\
&\geq \gamma.
\end{aligned}
$$

We conclude that $D_{i,i}^{(t)} \leq 1/\gamma$, for all $t \geq 0$.

$\square$

Writing the layer-wise update rule (3) recursively, we get the following formulation of $X^{(t)}$:

$$X^{(t+1)} = D^{(t)}A^{(t)}...D^{(0)}A^{(0)}X^{(0)}W^{(0)}...W^{(t)}.$$

We focus on the infinite product $\lim_{t \to \infty} D^{(t)}A^{(t)}...D^{(0)}A^{(0)}$. We first define a family of attention matrices as follows:

**Definition 6.** *Let $\epsilon > 0$. We define $\mathcal{A}_{\mathcal{G},\epsilon}$ to be the set of row-stochastic matrices satisfying the following conditions:*

1. *$\epsilon \leq A_{ij} \leq 1$, if $(j,i) \in E(\mathcal{G})$,*

2. *$A_{ij} = 0$, if $(j,i) \notin E$.*

We also define

$$\mathcal{D}_{C_1,C_2} = \{\text{diag}(\mathbf{d}) : C_1 \leq_{\text{ew}} \mathbf{d} \leq_{\text{ew}} C_2\}. \tag{21}$$

and subsequently,

$$\mathcal{M}_{\mathcal{G},\epsilon,C_1,C_2} = \{DA : D \in \mathcal{D}_{C_1,C_2}, A \in \mathcal{A}_{\mathcal{G},\epsilon}\}. \tag{22}$$

To study the infinite product of matrices from $\mathcal{M}_{\mathcal{G},\epsilon,C_1,C_2}$, we introduce some necessary concepts:

**Definition 7.** *A non-negative matrix $M$ is called* row allowable *if it has a positive entry in each of its rows.*

**Definition 8.** *Consider a sequence of row allowable matrices $\{M^{(t)}\}_{t=0}^{\infty}$. Define the partial product*

$$P^{(t)} := M^{(t)}...M^{(0)}.$$

*We say that the sequence $\{P^{(t)}\}_{t=0}^{\infty}$ is ergodic if there exists a sequence of positive rank one matrices $\{S^{(t)}\}_{t=0}^{\infty}$ such that*

$$\lim_{t\to\infty} \frac{P_{ij}^{(t)}}{S_{ij}^{(t)}} = 1, \quad \forall i, j \in [N]. \tag{23}$$

Our goal is to show the following key result:

**Lemma 7.** *Let $\epsilon, C_1, C_2 > 0$. Given $\{D^{(t)}A^{(t)}\}_{t=0}^{\infty}$ in $\mathcal{M}_{\mathcal{G},\epsilon,C_1,C_2}$. The sequence of partial products $\{P^{(t)}\}_{t=0}^{\infty}$ is ergodic.*

For that, we will make use of the following result:

**Lemma 8** (Corollary 5.1 [18]). *Consider a sequence of row allowable matrices $\{M^{(t)}\}_{t=0}^{\infty}$. Let $a_t$ and $b_t$ be the smallest and largest entries in $M^{(t)}$, respectively. If $\sum_{t=0}^{\infty} \frac{a_t}{b_t} = \infty$, then $\{P^{(t)}\}_{t=0}^{\infty}$ is ergodic.*

We cannot directly apply Lemma 8 to $D^{(t)}A^{(t)}$ yet unless in the special case of full bidirectional attention such as the one used in BERT i.e. $\mathcal{G}$ is an undirected, complete graph, as $a_t$ could equal zero due to the sparsity pattern of attention $\mathcal{G}$. In order to make use of the above result, we first need to show that long products of $P^{(t)} := D^{(t)}A^{(t)}$ will eventually become strictly positive for strongly connected $\mathcal{G}$. For $t_0 \leq t_1$, we denote

$$P^{(t_1:t_0)} = P^{(t_1)} \ldots P^{(t_0)}.$$

**Lemma 9.** *Under* **A1***, if $\mathcal{G}$ is strongly connected, there exist $T \in \mathbb{N}$ and $C_3, C_4 > 0$ such that for all $t \geq 0$,*

$$C_3 \leq P_{i,j}^{(t+T:t)} \leq C_4, \quad \forall 1 \leq i, j \leq N.$$

*Proof.* Note that by **A1** and the strong connectivity of $\mathcal{G}$, there exists $T \in \mathbb{N}$, e.g. the diameter of $\mathcal{G}$, and $\epsilon, C_1, C_2 > 0$ such that for all $t \geq 0$,

$$(\epsilon C_1)^T \leq P_{i,j}^{(t+T:t)} \leq C_2^T, \quad \forall 1 \leq i, j \leq N.$$

$\square$

Then we apply Lemma 8 to $\{P^{(k+1)T:T}\}_{k=0}^{\infty}$ and concludes Lemma 7.

As a result, from (23) we see that any $P^{(t)}$ can now be written in the following form:

$$P^{(t)} = u^{(t)}(v^{(t)})^{\top} + E^{(t)},$$

where $u^{(t)}, v^{(t)}$ are positive vectors such that $u^{(t)}(v^{(t)})^{\top} = S^{(t)}$. Without loss of generality, assume $\|v^{(t)}\|_2 = 1$.

Then it follow from (23) that

**Lemma 10.**

$$\lim_{t\to\infty} \frac{E_{ij}^{(t)}}{u_i^{(t)}} = 0, \quad \forall i, j \in [N].$$

*Proof.* Given

$$\left| v_j^{(t)} \frac{E_{ij}^{(t)}}{u_i^{(t)} v_j^{(t)}} \right| \leq \left| \frac{E_{ij}^{(t)}}{u_i^{(t)} v_j^{(t)}} \right|$$

where by (23), since also

$$\lim_{t \to \infty} \left| \frac{E_{ij}^{(t)}}{u_i^{(t)} v_j^{(t)}} \right| = 0 \,,$$

We get that

$$\lim_{t \to \infty} \left| \frac{E_{ij}^{(t)}}{u_i^{(t)}} \right| = 0 \,,$$

and hence

$$\lim_{t \to \infty} \frac{E_{ij}^{(t)}}{u_i^{(t)}} = 0 \,.$$

$\square$

Thus the 2-norm of the $i^{th}$ token of $X^{(t+1)}$, $\|X_{i,:}^{(t+1)}\|_2$ is

$$\|P_{i,:}^{(t)} X^{(0)} \tilde{W}^{(t)}\|_2 = \|u_i^{(t)} (v^{(t)})^\top X^{(0)} \tilde{W}^{(t)} + E_{i,:}^{(t)} X^{(0)} \tilde{W}^{(t)}\|_2 = 1 \,, \forall i \in [N]. \tag{24}$$

Then by Lemma 10,

$$\lim_{t \to \infty} \left\| (v^{(t)})^\top X^{(0)} \tilde{W}^{(t)} + \frac{E_{i,:}^{(t)}}{u_i^{(t)}} X^{(0)} \tilde{W}^{(t)} \right\|_2 = \lim_{t \to \infty} \left\| (v^{(t)})^\top X^{(0)} \tilde{W}^{(t)} \right\|_2 = \lim_{t \to \infty} \frac{1}{u_i^{(t)}} \,.$$

Note that

$$\min_{\|v\|_2 = 1} \left\| v^\top X^{(0)} \tilde{W}^{(t)} \right\|_2 = \sigma_{\min}((X^{(0)} \tilde{W}^{(t)})^\top) \,,$$

where $\sigma_{\min}(\cdot)$ denotes the smallest singular value. Since we assume $X^{(0)}$ has full rank, and $\tilde{W}^{(t)}$ is orthogonal

$$\lim_{t \to \infty} \left\| (v^{(t)})^\top X^{(0)} \tilde{W}^{(t)} \right\|_2 \geq \sigma_{\min}((X^{(0)} \tilde{W}^{(t)})^\top) > 0 \,.$$

Then given $\forall i \in [N]$,

$$\lim_{t \to \infty} \left\| (v^{(t)})^\top X^{(0)} \tilde{W}^{(t)} \right\|_2 = \lim_{t \to \infty} \frac{1}{u_i^{(t)}} \,, \forall i \in [N] \,,$$

as a result,

$$\lim_{t \to \infty} u_i^{(t)} = \lim_{t \to \infty} u_k^{(t)} < \infty \,, \forall i, k \in [N] \,. \tag{25}$$

This together with Lemma 10 implies that

$$\lim_{t \to \infty} E_{ij}^{(t)} = 0, \forall i \in [N], j \in [d] \,,$$

and

$$\lim_{t \to \infty} \mu(X^{(t)}) = \lim_{t \to \infty} \|B u^{(t)} (v^{(t)})^\top X^{(0)} \tilde{W}^{(t)} + B E^{(t)} X^{(0)} \tilde{W}^{(t)}\|_F = 0 \,. \tag{26}$$

### C.1.1 Exponential convergence rate

Having established the convergence in (26), we then establish the exponential convergence rate.

We define that

$$\alpha^{(t)} := \min_{i,j\in[N]} \langle X_{i,:}^{(t)}, X_{j,:}^{(t)}\rangle.$$

Similar to the derivation of (25), note that

$$\max_{\|v\|_2=1} \left\| v^\top X^{(0)} \tilde{W}^{(t)} \right\|_2 = \sigma_{\max}((X^{(0)}\tilde{W}^{(t)})^\top),$$

it follows that

$$\lim_{t\to\infty} \left\| (v^{(t)})^\top X^{(0)} \tilde{W}^{(t)} \right\|_2 \le \sigma_{\max}((X^{(0)}\tilde{W}^{(t)})^\top) < \infty.$$

Since $\forall i \in [N]$,

$$\lim_{t\to\infty} \left\| (v^{(t)})^\top X^{(0)} \tilde{W}^{(t)} \right\|_2 = \lim_{t\to\infty} \frac{1}{u_i^{(t)}}, \forall i \in [N]$$

as a result,

$$\lim_{t\to\infty} u_i^{(t)} = \lim_{t\to\infty} u_k^{(t)} > 0, \forall i, k \in [N],$$

meaning that $\lim_{t\to\infty} X_{i,:}^{(t)} \in S^{d-1}$ for all $i \in [N]$ .

Then given the convergence established in (26), we get that $\lim_{t\to\infty} \alpha^{(t)} = 1$, from which there exists $t_0 \in \mathbb{N}$ such that $\alpha^{(t)} > 0, \forall t \ge t_0$. It follows that for $t \ge t_0$,

$$\phi^{(t+r)} = \min_{i,j\in[N]} \langle X_{i,:}^{(t+r)}, X_{j,:}^{(t+r)}\rangle = \langle X_{i^{(t+r)},:}^{(t+r)}, X_{j^{(t+r)},:}^{(t+r)}\rangle$$

$$\ge \langle \sum_{k_1=1}^N A_{i^{(t+r)},k_1}^{(t+r-1)} X_{k_1}^{(t+r-1)}, \sum_{l_1=1}^N A_{j^{(t+r)},l_1}^{(t+r-1)} X_{l_1}^{(t+r-1)}\rangle$$

$$\ge \sum_{(k_1,\dots,k_r)\in[N]^r} \sum_{(l_1,\dots,l_r)\in[N]^r} A_{i^{(t+r)},k_1}^{(t+r-1)}\dots A_{k_{r-1},k_r}^{(t)} A_{j^{(t+r)},l_1}^{(t+r-1)}\dots A_{l_{r-1},l_r}^{(t)} \langle X_{k_r}^{(t)}, X_{l_r}^{(t)}\rangle$$

$$\ge N\epsilon^{2r} + (1 - N\epsilon^{2r})\phi^{(t)} .$$

Thus

$$1 - \phi^{(t+r)} \le (1 - N\epsilon^{2r})(1 - \phi^{(t)}).$$

Writing recursively, we get that for $k \ge 0$,

$$1 - \phi^{(t_0+kr)} \le (1 - N\epsilon^{2r})^k (1 - \phi^{(t_0)}),$$

Let $C := 2/(1 - N\epsilon^{2r})^{t_0/r}$. Hence for all $t \ge 0$,

$$1 - \alpha^{(t)} \le C(1 - N\epsilon^{2r})^{\frac{t}{r}} . \tag{27}$$

Since by definition, $1 - \phi^{(t)} \ge 1 - \langle X_{i,:}^{(t)}, X_{j,:}^{(t)}\rangle \ge \|X_{i,:}^{(t)} - X_{j,:}^{(t)}\|_2^2/2$, it follows that

$$\mu(X^{(t)}) = \|X^{(t)} - \mathbf{1}\mathbf{1}^\top X^{(t)}/N\|_F = \sqrt{\sum_{i=1}^N \|X_{i,:}^{(t)} - \mathbf{1}^\top X^{(t)}/N\|_2^2}$$

$$= \sqrt{\frac{1}{2N}\sum_{i=1}^N \sum_{j=1}^N \|X_{i,:}^{(t)} - X_{j,:}^{(t)}\|_2^2}$$

$$\le C'(1 - N\epsilon^{2r})^{\frac{t}{2r}},$$

where $C' = \sqrt{CN}$, completing the proof.

## D  Proof of Corollary 1

First, since for $x \in \text{Conv}(X^{(t)})$, $W^{(t)\top} x$ always lies in $\mathbb{S}^{d-1}$, $D_{i,i}^{(t)} \geq 1$, for all $i \in [N], t \geq 0$.

For all $t \geq 0$, since $\phi^{(0)} \geq 0$ and $\mathcal{G}$ is quasi-strongly connected, we have that

$$
\begin{aligned}
\phi^{(t+r)} &= \min_{i,j \in [N]} \langle X_{i,:}^{(t+r)}, X_{j,:}^{(t+r)} \rangle = \langle X_{i^{(t+r)},:}^{(t+r)}, X_{j^{(t+r)},:}^{(t+r)} \rangle \\
&\geq \langle \sum_{k_1=1}^{N} A_{i^{(t+r)},k_1}^{(t+r-1)} X_{k_1}^{(t+r-1)} W^{(t+r-1)}, \sum_{l_1=1}^{N} A_{j^{(t+r)},l_1}^{(t+r-1)} X_{l_1}^{(t+r-1)} W^{(t+r-1)} \rangle \\
&= \langle \sum_{k_1=1}^{N} A_{i^{(t+r)},k_1}^{(t+r-1)} X_{k_1}^{(t+r-1)}, \sum_{l_1=1}^{N} A_{j^{(t+r)},l_1}^{(t+r-1)} X_{l_1}^{(t+r-1)} \rangle \\
&\geq \sum_{(k_1,\ldots,k_r) \in [N]^r} \sum_{(l_1,\ldots,l_r) \in [N]^r} A_{i^{(t+r)},k_1}^{(t+r-1)} \ldots A_{k_{r-1},k_r}^{(t)} A_{j^{(t+r)},l_1}^{(t+r-1)} \ldots A_{l_{r-1},l_r}^{(t)} \langle X_{k_r}^{(t)}, X_{l_r}^{(t)} \rangle \\
&\geq \epsilon^{2r} + (1 - \epsilon^{2r}) \phi^{(t)}.
\end{aligned}
$$

Thus

$$
1 - \phi^{(t+r)} \leq (1 - \epsilon^{2r})(1 - \phi^{(t)}).
$$

Writing recursively, we get that for $k \geq 0$,

$$
1 - \phi^{(kr)} \leq (1 - \epsilon^{2r})^k (1 - \phi^{(0)}),
$$

Let $C := 1/(1 - \epsilon^{2r})$. Hence for all $t \geq 0$,

$$
1 - \phi^{(t)} \leq C(1 - \epsilon^{2r})^{\frac{t}{r}}. \tag{28}
$$

Since by definition, $1 - \phi^{(t)} \geq 1 - \langle X_{i,:}^{(t)}, X_{j,:}^{(t)} \rangle \geq \|X_{i,:}^{(t)} - X_{j,:}^{(t)}\|_2^2 / 2$, it follows that

$$
\begin{aligned}
\mu(X^{(t)}) &= \|X^{(t)} - \mathbf{1}\mathbf{1}^\top X^{(t)} / N\|_F = \sqrt{\sum_{i=1}^{N} \|X_{i,:}^{(t)} - \mathbf{1}^\top X^{(t)} / N\|_2^2} \\
&= \sqrt{\frac{1}{2N} \sum_{i=1}^{N} \sum_{j=1}^{N} \|X_{i,:}^{(t)} - X_{j,:}^{(t)}\|_2^2} \\
&\leq C'(1 - \epsilon^{2r})^{\frac{t}{2r}},
\end{aligned}
$$

where $C' = \sqrt{CN}$, completing the proof.

## E  Detailed analysis of the illustrative example in Section 4.3.1

**First token**  Given that

$$
\frac{X_{1,2}^{(t+1)}}{X_{1,1}^{(t+1)}} = \frac{X_{1,2}^{(t)}}{X_{1,1}^{(t)}} + w,
$$

we get that

$$
\frac{X_{1,2}^{(t+1)}}{X_{1,1}^{(t+1)}} = \frac{X_{1,2}^{(0)}}{X_{1,1}^{(0)}} + wt.
$$

and thus

$$
\lim_{t \to \infty} \frac{X_{1,2}^{(t)}}{X_{1,1}^{(t)}} = \infty.
$$

Since the sign of $X_{1,1}^{(t)}$ is invariant, depending on its initial sign, we get that

$$\lim_{t\to\infty} X_{1,:}^{(t)} = \begin{cases} [0,1] & \text{if } X_{1,1}^{(0)} > 0 \\ [0,-1] & \text{if } X_{1,1}^{(0)} < 0 \end{cases}.$$

**Second token**  Assume that the first token converges to $[0,1]$. Then

$$\frac{X_{2,2}^{(t+1)}}{X_{2,1}^{(t+1)}} = \frac{X_{2,2}^{(t)}}{X_{2,1}^{(t)}} + w + \frac{1}{X_{2,1}^{(t)}} := g\left(\frac{X_{2,2}^{(t)}}{X_{2,1}^{(t)}}\right),$$

Solving $\frac{X_{2,2}^{(t+1)}}{X_{2,1}^{(t+1)}} = \frac{X_{2,2}^{(t)}}{X_{2,1}^{(t)}}$ gives $X_{2,1} = -1/w$ and

$$\begin{cases} \frac{X_{2,2}^{(t+1)}}{X_{2,1}^{(t+1)}} > \frac{X_{2,2}^{(t)}}{X_{2,1}^{(t)}} & \text{if } X_{2,1} < -1/w \\ \frac{X_{2,2}^{(t+1)}}{X_{2,1}^{(t+1)}} < \frac{X_{2,2}^{(t)}}{X_{2,1}^{(t)}} & \text{otherwise}. \end{cases} \tag{29}$$

To show the convergence, we utilize the following useful property of $g(\cdot)$:

**Lemma 11.** *$g$ is non-decreasing in $X_{2,2}/X_{2,1}$.*

*Proof.* Let $y = X_{2,2}/X_{2,1}$. Given the identity that

$$\left(\frac{1}{X_{2,1}^{(t)}}\right)^2 = y^2 + 1,$$

we get that

$$2\frac{\partial\left(\frac{1}{X_{2,1}}\right)}{\partial y}\frac{1}{X_{2,1}} = 2y.$$

Hence

$$\frac{\partial\left(\frac{1}{X_{2,1}}\right)}{\partial y} = X_{2,1}y = X_{2,2},$$

which means that

$$\frac{\partial g}{\partial y} = 1 + X_{2,1} \geq 0.$$

$\square$

Given $X_{2,2}/X_{2,1}$ is invariant at $X_{2,1} = -1/w$ (symmetric points $A$ and $B$ in Fig. 5), together with Lemma 11, they imply the invariance of the 4 segments on $\mathbb{S}^1$, as shown in Fig. 5. Furthermore, (29) shows which point each region would converge to. Thus it suffices to choose $X_{2,:}^{(0)}$ to be on the dark blue or orange segment.

# F   Proof of Theorem 3

## F.1   Finding the equilibria

**Step 1**  Without loss of generality, we set that $W_K^{(t)} = W_Q^{(t)} = \mathbf{0}$, for all $t \geq 0$. The system of equations corresponds to the equilibrium condition for the attention dynamics with LayerNorm defined in (3) when the attention mask $\mathcal{G}$ is causal is as follows:

$$\begin{cases} X_{1,:} = F_1(X) = d_1 X_{1,:} W_V \\ X_{2,:} = F_2(X) = d_2 \left(\frac{1}{2}X_{1,:}W_V + \frac{1}{2}X_{2,:}W_V\right) \\ \vdots \\ X_{N,:} = F_N(X) = d_N \left(\frac{1}{N}X_{1,:}W_V + ... + \frac{1}{N}X_{N,:}W_V\right) \end{cases}$$

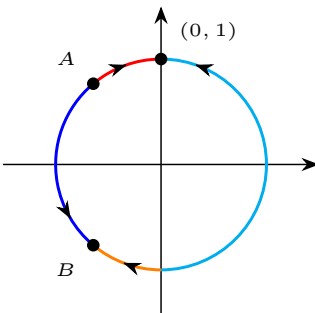

Figure 5: Convergence analysis of the second token in $d = 2$.

where $d_i = i/\|\sum_{j=1}^{i} X_{j,:} W_V\|_2$. We define that $\beta_i = d_i/i$ for $i \in [N]$. Then the above system equation becomes

$$\begin{cases} X_{1,:} = \beta_1 X_{1,:} W_V \\ X_{2,:} = \beta_2 \left( \frac{1}{\beta_1} X_{1,:} + X_{2,:} W_V \right) \\ \vdots \\ X_{N,:} = \beta_N \left( \frac{1}{\beta_{N-1}} X_{N-1,:} + X_{N,:} W_V \right) \end{cases}$$

after substitution. Then for $i \in [N-1]$,

$$X_{i+1,:} = \beta_{i+1} \left( \frac{1}{\beta_i} X_{i,:} + X_{i+1,:} W_V \right) , \tag{30}$$

Let $k \in [N]$. If $\beta_1 = ... = \beta_k = 1$, then for $i \in [k-1]$,

$$X_{i+1,:} = X_{i,:} + X_{i+1,:} W_V , \tag{31}$$

Then if $X_{K,:}$ is a generalized eigenvector of rank $k$ corresponding to $W_V$ with eigenvalue 1, then from (31) that $\{X_{K,:}, ..., X_{1,:}\}$ is the Jordan chain generated by $X_{K,:}$, which is known to be linearly independent.

Hence we set $W_V$ to be a matrix with a generalized eigenvector of rank $k$ corresponding to eigenvalue 1. For example,

1. For $k = 1$, $W_V$ can be set as $I_d$;

2. For $k = 2$, $W_V$ can be set as

$$W_V^{(t)} = \begin{bmatrix} 1 & 0 & 0 & ... & 0 & 0 \\ 0 & 1 & 0 & ... & 0 & 0 \\ \vdots & \vdots & \vdots & ... & \vdots & \vdots \\ 0 & 0 & 0 & ... & 0 & 0 \\ 0 & 0 & 0 & ... & 1 & w \\ 0 & 0 & 0 & ... & 0 & 1 \end{bmatrix} \quad \forall t \geq 0 .$$

3. For $k = 3$, $W_V$ can be set as

$$W_V^{(t)} = \begin{bmatrix} 1 & 0 & 0 & ... & 0 & 0 \\ 0 & 1 & 0 & ... & 0 & 0 \\ \vdots & \vdots & \vdots & ... & \vdots & \vdots \\ 0 & 0 & 0 & ... & w & 0 \\ 0 & 0 & 0 & ... & 1 & w \\ 0 & 0 & 0 & ... & 0 & 1 \end{bmatrix} \quad \forall t \geq 0 .$$

4. For $k = N$, $W_V$ can be set as

$$W_V^{(t)} = \begin{bmatrix} 1 & w & 0 & ... & 0 & 0 \\ 0 & 1 & w & ... & 0 & 0 \\ \vdots & \vdots & \vdots & ... & \vdots & \vdots \\ 0 & 0 & 0 & ... & w & 0 \\ 0 & 0 & 0 & ... & 1 & w \\ 0 & 0 & 0 & ... & 0 & 1 \end{bmatrix} \quad \forall t \geq 0. \tag{32}$$

For our own purpose, we consider $k = N = d$. Then to solve for equilibrium defined by the condition in (31), for $i \in [N-1]$,

$$X_{i+1,:} \begin{bmatrix} 0 & -w & 0 & ... & 0 & 0 \\ 0 & 0 & -w & ... & 0 & 0 \\ \vdots & \vdots & \vdots & ... & \vdots & \vdots \\ 0 & 0 & 0 & ... & -w & 0 \\ 0 & 0 & 0 & ... & 0 & -w \\ 0 & 0 & 0 & ... & 0 & 0 \end{bmatrix} = X_{i,:}\,,$$

which means that

$$[0, -wX_{i+1,1}, -wX_{i+1,2}, ..., -wX_{i+1,d-1}] = [X_{i,1}, X_{i,2}, X_{i,3}, ..., X_{i,d}]. \tag{33}$$

Then setting $X_{1,:} = [0, 0, ..., 0, 1]$, iteratively solving (31), we get that an equilibrium satisfying (30) is as follows:

$$\begin{cases} X_{1,:} = [0, 0, ..., 0, 0, 1] \\ X_{2,:} = \left[0, 0, ..., 0, -\frac{1}{w}, -\sqrt{1 - \frac{1}{w^2}}\right] \\ X_{3,:} = \left[0, 0, ..., \frac{1}{w^2}, \sqrt{\frac{1}{w^2} - \frac{1}{w^4}}, \sqrt{1 - \frac{1}{w^2}}\right] \\ \vdots \\ X_{N,:} = (-1)^{N-1} \begin{bmatrix} \frac{1}{w^{N-1}} \\ \sqrt{\frac{1}{w^{2(N-2)}} - \frac{1}{w^{2(N-1)}}} \\ ... \\ \sqrt{\frac{1}{w^4} - \frac{1}{w^6}} \\ \sqrt{\frac{1}{w^2} - \frac{1}{w^4}} \\ \sqrt{1 - \frac{1}{w^2}} \end{bmatrix}^{\top} \end{cases}, \tag{34}$$

which is valid if $w > 1$. Since from (31), there is a free coordinate, in fact there a $2^N$ many such equilibria. Hence there exists $\{W_K^{(t)}, W_Q^{(t)}, W_V^{(t)}\}_{t=0}^{\infty}$ such that the corresponding attention dynamics with LayerNorm has an equilibrium with rank $k = N$.

**Step 2** To show that there are equilibria of any rank between 1 and full co-existing for a corresponding dynamics given the same set of $\{W_K^{(t)}, W_Q^{(t)}, W_V^{(t)}\}_{t=0}^{\infty}$, consider $W_V^{(t)}$ to be the case of (32). Then notice that to show there exists equilibrium of $k$ beyond the full rank one as derived in Step 1, notice the following:

We can set the first $N - k + 1$ tokens to be $[0, ..., 0, 1]$, which still satisfies the equilibrium condition. This implies that $\beta_{N-k+1} = 1/(N - K + 1)$. Then recall (30):

$$X_{i+1,:} = \beta_{i+1} \left(\frac{1}{\beta_i} X_{i,:} + X_{i+1,:} W_V\right)$$

Let $\beta_{N-k+2} = 1$, we get that

$$X_{N-k+2,:}(I - W) = [0, 0, ..., 0, N - k + 1],$$

which yields

$$X_{N-k+2,:} = \left[ \mathbf{0}_{d-2}, -\frac{N-k+1}{w}, \pm\sqrt{1 - \frac{(N-k+1)^2}{w^2}} \right].$$

Continue the process by setting $\beta_i = 1$ for $i \geq N - k + 2$, we get a rank $k$ equilibrium as desired (there are $2^k$ such equilibria), as long as $w \geq N$.

**Step 3** Finally to ensure $\{W_K^{(t)}, W_Q^{(t)}, W_V^{(t)}\}_{t=0}^{\infty}$ satisfies **A2**-**A3**:

- We see that $\{W_K^{(t)}, W_Q^{(t)}\}_{t=0}^{\infty}$ satisfies **A2** by construction;

- Since this is a dynamics with LayerNorm, applying $W_V^{(t)}$ is in fact equivalent to applying $\hat{W}_V = W_V / \|W_V\|_2$ under the dynamics, where the latter has 2-norm at most 1 and thus using $\hat{W}_V$ satisfies **A3**.

### F.2 Anisotropy of the full rank equilibria

To show that the full rank equilibria $X^*$ lie in a narrow region, we make use of the notion stable rank:

$$\mathrm{SRank}(X) = \frac{\|X\|_F^2}{\|X\|_2^2}.$$

We would like to show that for any $\delta > 0$, weights can be chosen such that

$$1 \leq \mathrm{SRank}(X^*) \leq (1 + \delta) + O(1/N).$$

Since for any matrix $M$, $\|M\|_F^2 \geq \|M\|_2^2$, the first inequality trivially holds. To show the second inequality, consider the equilibrium in (34). Then direct calculation yields that

$$\|X^*\|_F^2 = \mathrm{Tr}(X^*(X^*)^\top) = N.$$

Since by definition,

$$\|X^*\|_2 = \sup_{\|v\|_2=1} \|X^*v\|_2,$$

and thus any $\|X^*v\|_2$ such that $v \in \mathbb{R}^N$, $\|v\|_2 = 1$ serves as a lower bound for $\|X\|_2$.

We consider $v = [0, 0, ..., 0, 1]^\top \in \mathbb{R}^d$. Then it follows that

$$\|X^*\|_2^2 \geq 1 + (N-1)\left(1 - \frac{1}{w^2}\right),$$

which implies that

$$\mathrm{SRank}(X^*) = \frac{\|X^*\|_F^2}{\|X^*\|_2^2} \leq \frac{N}{N - \frac{N-1}{w^2}}$$

Hence for any $\delta > 0$, it suffices to choose $w \geq 1$ such that

$$w \geq \sqrt{\frac{1}{\delta} + 1}.$$

### F.3 Convergence analysis

The convergence analysis for the case $d = 2$ can be found in Appendix E. Here, we deal with the general case where $d > 2$.

To show that tokens do not converge to an rank one subspace, it suffices to show that given certain conditions on the initial input $X^{(0)}$, there are invariant space of the tokens that is at least rank 2. To that end, we analyze the convergence of the first two tokens.

**First token** Given that

$$\frac{X_{1,2}^{(t+1)}}{X_{1,1}^{(t+1)}} = \frac{X_{1,2}^{(t)}}{X_{1,1}^{(t)}} + w\,,$$

we get that

$$\frac{X_{1,2}^{(t+1)}}{X_{1,1}^{(t+1)}} = \frac{X_{1,2}^{(0)}}{X_{1,1}^{(0)}} + wt\,.$$

Let $\frac{X_{1,2}^{(0)}}{X_{1,1}^{(0)}} = r_2 w$, for $r_2 \in \mathbb{R}$. It follows from above that

$$\frac{X_{1,2}^{(t)}}{X_{1,1}^{(t)}} = (r_2 + t)w\,. \tag{35}$$

Then for all $3 \le i \le d$,

$$\frac{X_{1,i}^{(t+1)}}{X_{1,1}^{(t+1)}} = w\frac{X_{1,i-1}^{(t)}}{X_{1,1}^{(t)}} + \frac{X_{1,i}^{(t)}}{X_{1,1}^{(t)}}\,, \tag{36}$$

setting

$$\frac{X_{1,i}^{(0)}}{X_{1,1}^{(0)}} = r_i w^{i-1}\,,$$

where $r_i \in \mathbb{R}$, and substituting from the base case (35) to (36) recursively, we get that

$$\frac{X_{1,i}^{(t)}}{X_{1,1}^{(t)}} = w^{i-1}\left( r_i + r_{i-1}t + r_{i-2}\sum_{j_1=0}^{t-1} j_1 + r_{i-3}\sum_{j_2=0}^{t-1}\sum_{j_1=0}^{j_2} j_1 + ... + \right.$$
$$\left. + r_2 \sum_{j_{i-3}=0}^{t-1} ... \sum_{j_2=0}^{j_3}\sum_{j_1=0}^{j_2} j_1 ... + \sum_{j_{i-2}=0}^{t-1} ... \sum_{j_2=0}^{j_3}\sum_{j_1=0}^{j_2} j_1 \right)\,, \tag{37}$$

which implies that

$$\lim_{t\to\infty} \frac{X_{1,1}^{(t+1)}}{X_{1,i}^{(t+1)}}\frac{X_{1,i}^{(t)}}{X_{1,1}^{(t)}} = 1\,. \tag{38}$$

Further, for $i \ge 3$

$$\frac{X_{1,i}^{(t+1)}}{X_{1,i-1}^{(t+1)}} = \frac{wX_{1,i-1}^{(t)} + X_{1,i}^{(t)}}{wX_{1,i-2}^{(t)} + X_{1,i-1}^{(t)}} = \frac{X_{1,1}^{(t+1)}}{X_{1,i-1}^{(t+1)}}\frac{X_{1,i-1}^{(t)}}{X_{1,1}^{(t)}}\left( w + \frac{X_{1,i}^{(t)}}{X_{1,i-1}^{(t)}} \right)\,. \tag{39}$$

Then (38) and (39) yields that

$$\lim_{t\to\infty} \frac{X_{1,i}^{(t)}}{X_{1,i-1}^{(t)}} = \infty\,,$$

which let us conclude that if $X_{1,1}^{(0)} > 0$, the first token would converge to $[0, 0, ..., 0, 1]$.

**Second token** Assume that the first token has converged to $[0, 0, ..., 0, 1]$. Similar to the first token, we can show that for $2 \le i \le d - 1$, it holds that

$$\lim_{t\to\infty} \frac{X_{2,i}^{(t)}}{X_{2,i-1}^{(t)}} = \infty\,,$$

and

$$\frac{X_{2,i}^{(t)}}{X_{2,1}^{(t)}} = w^{i-1}\left(r_i + r_{i-1}t + r_{i-2}\sum_{j_1=0}^{t-1}j_1 + r_{i-3}\sum_{j_2=0}^{t-1}\sum_{j_1=0}^{j_2}j_1 + ...+\right.$$

$$\left. + r_2\sum_{j_{i-3}=0}^{t-1}...\sum_{j_2=0}^{j_3}\sum_{j_1=0}^{j_2}j_1... + \sum_{j_{i-2}=0}^{t-1}...\sum_{j_2=0}^{j_3}\sum_{j_1=0}^{j_2}j_1\right),\qquad (40)$$

by setting $X_{2,i}^{(0)}/X_{2,1}^{(0)} = r_i w^{i-1}$, where $r_i > 0$. Then note that

$$\frac{X_{2,d}^{(t+1)}}{X_{2,d-1}^{(t+1)}} = \frac{wX_{2,d-1}^{(t)} + X_{2,d}^{(t)} + 1}{wX_{2,d-1}^{(t)} + X_{2,d}^{(t)}} = \frac{X_{2,1}^{(t+1)}}{X_{2,d-1}^{(t+1)}}\frac{X_{2,d-1}^{(t)}}{X_{2,1}^{(t)}}\left(w + \frac{X_{2,d}^{(t)}}{X_{2,d-1}^{(t)}} + \frac{1}{X_{2,d-1}^{(t)}}\right)$$

$$:= h\left(\frac{X_{2,3}^{(t)}}{X_{2,2}^{(t)}}, t, r_0, ..., r_{d-1}\right).$$

We will show a useful monotone property of $h(\cdot)$.

**Lemma 12.** *Fix $t, r_2, ..., r_{d-1} > 0$, $h$ is non-decreasing in $X_{2,d}/X_{2,d-1}$.*

*Proof.* Let $y = X_{2,d}/X_{2,d-1}$. Given the identity that

$$\left(\frac{1}{X_{2,d-1}^{(t)}}\right)^2 = y^2 + 1 + \frac{X_{2,1}^2 + ... + X_{2,d-2}^2}{X_{2,d-1}^2} := y^2 + 1 + f(r_2, ..., r_d),$$

we get that

$$2\frac{\partial\left(\frac{1}{X_{2,d-1}}\right)}{\partial y}\frac{1}{X_{2,d-1}} = 2y.$$

Hence

$$\frac{\partial\left(\frac{1}{X_{2,d-1}}\right)}{\partial y} = X_{2,d-1}y = X_{2,d},$$

which means that

$$\frac{\partial h}{\partial y} = \frac{X_{2,1}^{(t+1)}}{X_{2,d-1}^{(t+1)}}\frac{X_{2,d-1}^{(t)}}{X_{2,1}^{(t)}}(1 + X_{2,d}) \geq 0.$$

$\square$

With this, we will show the following claim proving the invariance of the subspace of $X_{2,d-1} \leq -1/w$ given some initial conditions:

**Lemma 13.** *Suppose $X_{2,i}^{(0)} < 0$ for $i \in [d-2]$, $X_{2,d-1}^{(0)} \leq -1/w$ such that*

$$\frac{1}{w^2}\left(\frac{r_{d-2}}{r_{d-1}}\right)^2 + \frac{1}{w^4}\left(\frac{r_{d-3}}{r_{d-1}}\right)^2 + ... + \frac{1}{w^{2(d-3)}}\left(\frac{r_2}{r_{d-1}}\right)^2 + \frac{1}{w^{2(d-2)}}\left(\frac{1}{r_{d-1}}\right)^2 \leq \sqrt{w^2 - 1}, \quad (41)$$

*then $X_{2,d-1}^{(t)} \leq -1/w$ for all $t \geq 0$.*

*Proof.* When $X_{2,d-1}^{(t)} = -1/w$, and given the conditions on $r_2, ..., r_{d-1}$ in (41),

$$\frac{X_{2,d}^{(t)}}{X_{2,d-1}^{(t)}} = \pm\sqrt{w^2 - 1 - \frac{\left(X_{2,1}^{(t)}\right)^2 + ... + \left(X_{2,d-2}^{(t)}\right)^2}{\left(X_{2,d-1}^{(t)}\right)^2}}$$

$$= \pm\sqrt{w^2 - 1 - f(t, r_1, ..., r_{d-2})}$$

is well defined, where $f(\cdot)$ is monotone non-increasing in $t$ for fixed $r_1, ..., r_{d-2} > 0$. We denote

$$\bar{y}^{(t)} = \sqrt{w^2 - 1 - f(t, r_1, ..., r_{d-2})}\,.$$

Note that if $X_{2,d-1}^{(t)} \leq -1/w$, we have

$$\frac{X_{2,d}^{(t)}}{X_{2,d-1}^{(t)}} \in \left[-\bar{y}^{(t)}, \bar{y}^{(t)}\right]$$

then by Lemma 12 we get that

$$h\left(\frac{X_{2,d}^{(t)}}{X_{2,d-1}^{(t)}}, t, r_2, ..., r_{d-1}\right) \in \left[-h\left(\bar{y}^{(t)}, t, r_2, ..., r_d\right), h\left(\bar{y}^{(t)}, t, r_2, ..., r_d\right)\right]$$

and since by (40),

$$\frac{X_{2,1}^{(t+1)}}{X_{2,d-1}^{(t+1)}} \frac{X_{2,d-1}^{(t)}}{X_{2,1}^{(t)}} \leq 1\,,$$

it implies that

$$\left|h\left(\pm\bar{y}^{(t)}\right)\right| = \frac{X_{2,1}^{(t+1)}}{X_{2,d-1}^{(t+1)}} \frac{X_{2,d-1}^{(t)}}{X_{2,1}^{(t)}} \bar{y}^{(t)} \leq \bar{y}^{(t+1)}\,,$$

which means that

$$\frac{X_{2,d-1}^{(t+1)}}{X_{2,d}^{(t+1)}} \in \left[-\bar{y}^{(t+1)}, \bar{y}^{(t+1)}\right]$$

and thus $X_{2,d-1}^{(t)} \leq -1/w$ for $t \geq 0$. $\qquad\square$

This let us conclude that for a general class of sequences $X^{(0)}$, $X^{(t)}$ does not converge to a rank one subspace as $t \to \infty$.

### F.4  Further discussion

**Directionality of attention mask $\mathcal{G}$**   In the proof of Theorem 3, we would like to note that $\mathcal{G}$ being the causal graph gives a key structure to construct the counterexamples. A causal graph essentially introduces asymmetries among token interactions where some tokens can attend to some other tokens but not verse versa and we exploit such asymmetries in constructing the counterexamples. A similar construction thus does not naturally apply to the case where $\mathcal{G}$ is undirected, e.g. a complete graph.

As a result, we leave the following interesting question for future research about whether the directionality of graphs empowers the attention mechanism differently. We conjecture a negative answer based on preliminary simulations.

> *Does a similar equilibrium exist if one uses an undirected graphs $\mathcal{G}$ as the attention mask? If not, what differentiates the case between an undirected graph and a directed graph?*

## G  Experiments

Here we provide more details about numerical experiments presented in Section 5 and some additional experimental results. All models were implemented with PyTorch [29] and Transformers library [36].

**Compute**   We ran all of our experiments on CPUs.

**Licenses**

- Libraries
  - Wikipedia: MIT license
  - tranformers [36]: Apache License 2.0
- Pretrained models
  - BERT (bert-base-uncased) [11]: Apache License 2.0
  - GPT2 (openai-community/gpt2) [30]: MIT license
  - T5 (google-t5/t5-base) [31]: Apache License 2.0
  - ALBERT (albert/albert-base-v2) [24]: Apache License 2.0

## G.1 Deeper models

We provide the results for deeper models for the experiment presented in Fig. 2. Here, we conduct the same experiment in a 128 layer randomly initialized BERT with 12 heads and 768 hidden dimension using different attention masks. Similarly, in SANs, while different masks have different convergence rates, all of them suffer from exponential rank collapse. But as soon as we add in LayerNorm, $\mu(X^{(t)})$ no longer converges to 0, as $\mu(\cdot)$ does not show any decreasing trend even after 128 layers.

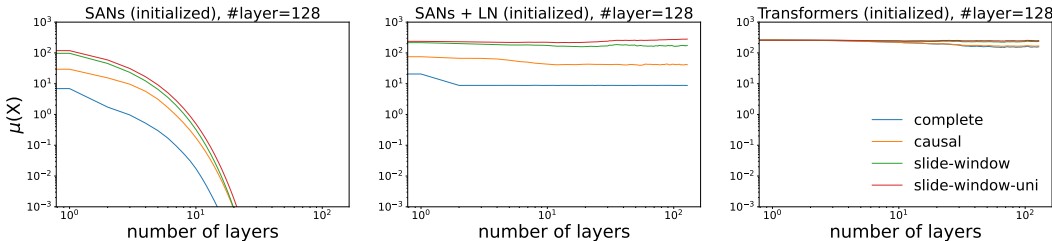

Figure 6: Evolution of $\mu(X^{(t)})$ (in log-log scale) as the number of layers increases in 128 layer models.

## G.2 The effect of the temperature term $d_{QK}$ in initialized transformers

The effect of the temperature term $d_{QK}$ in initialized transformers is shown in Fig. 7. Similar to the pretrained models, we see that smaller temperature terms alleviate the rate of rank collapse, and effect is more significant under global attention than under sparser masked attention, and in shallower layers than deeper layers.

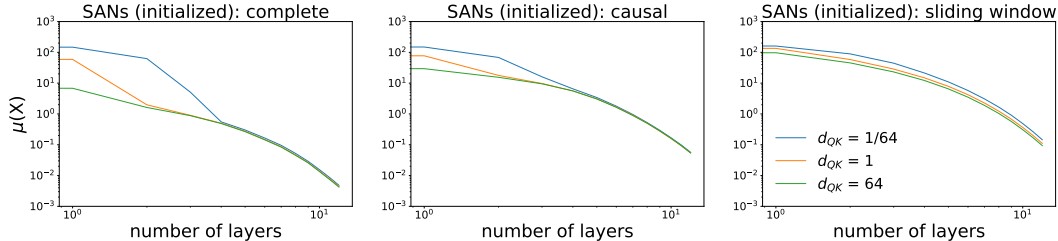

Figure 7: Evolution of $\mu(X^{(t)})$ (in log-log scale) as the number of layers increases in SANs with different $d_{QK}$ at initialization.

## G.3 The evolution of singular values of token representations in transformers

In this section, we investigate the full set of singular values of token representations $X^{(t)}$ in transformers. For a clear presentation, we randomly select 3 out of the 3000 sequence samples described in Section 5 and calculate all the singular values of $X^{(t)}$. The results in randomly initialized transformers are presented in Fig. 8.

We see that while the rank is full for $X^{(t)}$, there is usually one principal component that is much more dominant than the rest. As we have discussed in the main text, this is the anisotropic characteristic of token representations— there is a strong non-uniformity in different directions. Nonetheless, it is important to note that the "small" singular values are in fact not negligible here in the absolute sense—the minimal singular values are never close to zero, as Fig. 4 (middle) shows.

### G.4    Initial token geometry

We verify the initial token geometry in transformers. For a clear presentation, we randomly select 3 out of the 3000 sequence samples described in Section 5 and calculate all the pairwise cosine similarities between tokens. The results in randomly initialized transformers are presented in Fig. 9. We see that in particular in BERT and ALBERT, the initial pairwise cosine similarities are all non-negative.

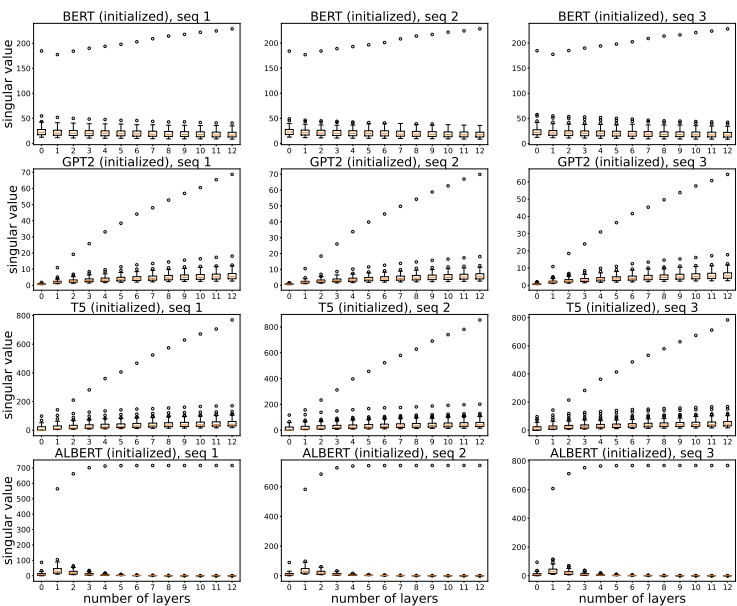

Figure 8: The full set of singular values of $X^{(t)}$ at each layer in initialized and pretrained transformers.

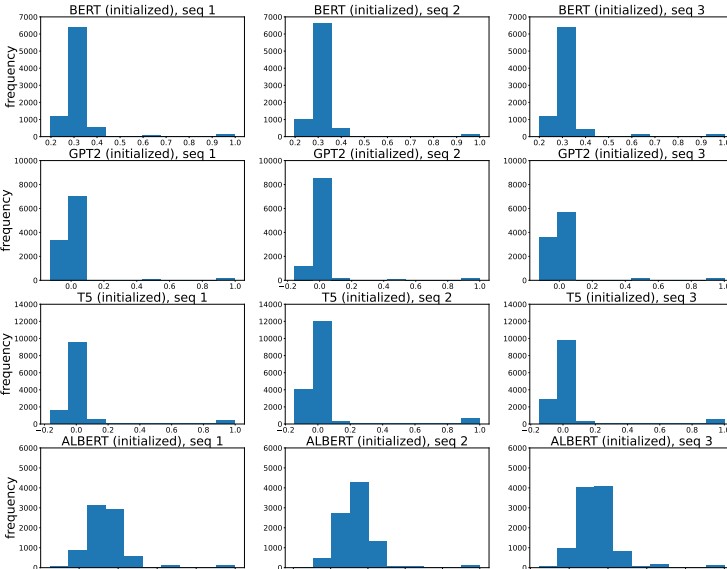

Figure 9: Pairwise cosine similarities between tokens calculated from the initial embeddings in randomly initialized and pretrained transformers, respectively.

