# OpenReview forum: "On the Role of Attention Masks and LayerNorm in Transformers"
_NeurIPS.cc/2024/Conference — NeurIPS 2024 poster_

### Official Review · Reviewer_ivrv · 2024-07-05

**Soundness:** 3
**Presentation:** 2
**Contribution:** 2
**Rating:** 4
**Confidence:** 4

**Summary:**

- This paper investigates the role of attention masks and layer normalization (LayerNorm) in mitigating the rank collapse issue in transformer models, and gets following conclusions:
	- A long as there is a token which all other tokens in the sequence can directly or indirectly attend to over a fixed number of layers, exponential rank collapse is guaranteed.
	- local attention mask or focus attention can slow down the collapse.
	- layernorm plays an important role in self-attention dynamics.
- This paper provides extensive analysis and numerical experiments on the self attention dynamics, Expressivity and Versatility.

**Strengths:**

- The paper provides a rigorous mathematical analysis of the role of attention masks and LayerNorm in transformers, contributing to a deeper understanding of these models' inner workings.
- This paper uses a graph-theoretic approach to analyze self-attention, so it is more general and can be extended to sparse attention and causal attention.
- This paper shows an interesting counterexample of collapse and shows that self-attention dynamics can process a rich set of sequences stablely.
- The paper supports its theoretical findings with numerical experiments.

**Weaknesses:**

- Application of the conclusion, I wonder can the findings in this paper guide some design choices in the future transformer models.
- And how the findings connect to (or explain) the strong performances of of existing LLMs? It claims that sparse attention can slow down collapse, why most existing LLMs do not use sparse attention?
- The numerical experiments are somewhat weak as the results are obtained from only 32 samples.

**Questions:**

I wonder what is the differences of self-attention dynamics between the causal attention and bidirectional attention? I cannot find clear analysis about this in the paper, since different self-attention methods are generalized with graph connections.

---

> ### Author Rebuttal · Authors · 2024-08-07
>
> We thank the reviewer for taking the time to review our paper and providing constructive feedback. In line with the reviewer’s suggestion, we have rerun all the numerical experiments, validating our theoretical findings on 3000 examples. The results are provided in the rebuttal pdf file. In what follows, we provide detailed responses to the rest of the comments raised by the reviewer.
>
> > Q1: Application of the conclusion, I wonder can the findings in this paper guide some design choices in the future transformer models.
>
> Thank you for the question. Here are some ideas along our theoretical results:
>
> - The graph-dependent rate of rank collapse established in the paper can be used as a principle for attention mask design. It is, however, important to also note that the design cannot be solely based on this result. One also needs to consider other implications of the mask, including its effects on universal approximation property [1]. We have discussed this remark in in line 178-184 of the manuscript.
>
> - Our result could also further explain the role and importance of LayerNorm in transformers. It clarifies a misconception popularized by a previous work [2] that “layer normalization plays no role” for rank collapse in transformers. We rigorously prove that layer normalization can in principle prevent rank collapse and stabilize token representations in transformers, enabling the model to utilize depth more efficiently.
>
> > Q2: How do the findings connect to (or explain) the strong performances of existing LLMs? It claims that sparse attention can slow down collapse, why do most existing LLMs not use sparse attention?
>
> - Existing implementations of LLMs utilize LayerNorm, which is in line with our results on the role of LayerNorm in preventing rank collapse. This could justify why this module is preserved in the design of pioneering LLMs and improved variants.
>
> - As we discussed in the response to Q1, rank collapse is not the only issue one needs to consider when designing the attention mask. One also needs to consider aspects such as the universal approximation power of masked attention. How to balance the trade-off for more powerful LLMs is a vital research direction which we are currently exploring.
>
> - In fact, while many existing LLMs do not use sparse attention, sparse attention is used in practice and is gaining popularity due to the demand for efficiency. For example, sparse attention was populated by Longformer [3] and OpenAI [4] and nowadays LLMs like Mistral 7B use sliding window attention (SWA) [4]. Other popular sparse attention models include, but not limited to BigBird [5], Recurrent Memory Transformers (RMTs) [6,7] and StreamingLLM [8].
>
> - Besides language tasks, sparse attention is also common in vision transformers. Examples of the use can be found in the following papers [9,10,11].
>
> We will update our manuscript for an improved literature review on sparse attention.
>
> > Q3: The numerical experiments are somewhat weak as the results are obtained from only 32 samples.
>
> Thank you for your constructive suggestion. Our original sample size 32 was chosen according to the experiments in Dong et al. [2]. In line with the reviewer’s suggestion, we validate our theoretical findings on 3000 examples instead. The results are provided in the rebuttal pdf file. They are still similar to the original results; however, as suggested by the reviewer, the larger sample size makes them more reliable.
>
> We appreciate your questions and comments very much. Please let us know if there are any further questions.
>
> -----------------
> **References**
>
> [1] Yun et al.  O(n) connections are expressive enough: Universal approximability of sparse transformers. In NeurIPS, 2020.
>
> [2] Dong et al. Attention is not all you need: pure attention loses rank doubly exponentially with depth. In ICML, 2020.
>
> [3] Beltagy et al. Longformer: The long-document transformer. 2020.
>
> [4] Child et al. Generating Long Sequences with Sparse Transformers. 2019.
>
> [5] Jiang et al. Mistral 7B. 2023.
>
> [6] Zaheer et al. Bigbird: Transformers for longer sequences. In NeurIPS, 2020.
>
> [7] Bulatov et al. Recurrent Memory Transformer. In NeurIPS, 2022.
>
> [8] Bulatov et al. Scaling Transformer to 1M tokens and beyond with RMT. 2024.
>
> [9] Xiao et al. Efficient Streaming Language Models with Attention Sinks. In ICLR, 2024.
>
> [10] Liu et al. Swin transformer: Hierarchical vision transformer using shifted windows. In ICCV, 2021.
>
> [11] Pan et al. Slide-Transformer: Hierarchical Vision Transformer with Local Self-Attention. In CVPR, 2023.
>
> [12] Hassani et al. Neighborhood Attention Transformer. In CVPR, 2023.

---

### Official Review · Reviewer_7spk · 2024-07-07

**Soundness:** 3
**Presentation:** 3
**Contribution:** 2
**Rating:** 6
**Confidence:** 4

**Summary:**

This paper theoretically studies the role of attention masks and layer norm in the convergence to the rank collapse degeneracy in Transformers, which are two architectural components that have previously been overlooked in studying the rank collapse phenomenon. The authors first define the problem through the lens of graph theory. They then show that self-attention with general mask structure leads to rank collapse, but that factors such as the “graph diameter” of the mask (related to locality of attention) and the uniformity of attention (related to the temperature). They continue to show that LayerNorm does not prevent rank collapse for general attention masks with orthogonal value matrices, but that counterexamples exist for other choices of value matrices.

**Strengths:**

1. The paper is mostly well written and clear.
2. The paper provides new insights into the rank collapse phenomenon. In particular, the graph theoretic formulations provides general conditions in terms of attention mask (e.g. giving a quasi-strongly connected graph) that lead to rank collapse.
3. Moreover, the paper shows a more nuanced analysis for the collapse possibilities when a more complete architecture is considered (in particular with layer normalisation) .
4. The paper also shows the impact of hyperparams like attention context length or temperature in causing/preventing rank collapse.

**Weaknesses:**

My concerns are presented in order of importance (to me).

1. My main concern is that the theoretical setting considered omits several other important architectural components which are used in practice. In particular skip connections (which are known to be effective in preventing rank collapse properties a lot, e.g. Dong et al 2020, Noci et al 2022 or He et al 2023), or positional encoders. The paper is motivated by the fact that rank collapse papers do not study standard archs with all components included, but the omission of such architectural components (especially skips) also means one could argue the paper does not meet its goal.
2. The experimental verification of the theory seems a bit unconvincing at the moment. For example, the effect of temperature seems to disappear after 5 or 6 layers in the non-sliding window settings, and there do not seem to be error bars in figs 2/3 which makes one wonder if the results are significant (are the error bars just smaller than what is visible?). Moreover, it would be good to consider depths of larger than 10, as modern deep transformers can be up to ~100 layers deep and the effects should be more pronounced at larger depths. Also, there doesn't seem to be a difference between the top row of Figure 2 vs the bottom. Finally, how do you pretrain a SAN (figure 2 bottom left) without skip connections as the rank collapse literature would tell you that such models are not trainable (Noci et al 2022).
3. Somewhat related to my first point, it is not clear in my reading that the results presented concerning the counterexamples with particular value weights provide new insights into the fundamental behaviours of transformers in practice or rather represent theoretical edge cases which are nice to know exist but never occur in practice. Some questions that could address this: do such value weights (or value weights with these properties) appear in trained Transformers, or do these theoretical insights generate new initialisations/parameterisations for value weights to train transformers, given that orthogonal weights will lead to rank collapse even with layernorm (Theorem 2).
4. I would also argue that the intuition for the given counterexamples is not particularly clear at present (though as I say most other parts of the paper are well presented). As I understand it, the mathematical argument is that the "center node" has a neuron that has 0 activation due to the layernorm, and as a result even when attention allows other tokens to see the center node's activations the zero activations are not being transfered to other tokens and this prevents collapse. This can be clearer e.g. in section 4.3.1, but as I say I have reservations about how fundamental/important this insight could be.

**Questions:**

1. For the results without LayerNorm (e.g. Theorem 1 or figure 2 left plots), it seems like the definition of mu(X) doesn't rule out that simply X converges to 0 in Frobenius norm, so that the activations simply go to 0, as opposed to saying something about the different tokens becoming aligned (in terms of cosine similarity going to 1) which is what some previous works have described as rank collapse. Do different attention masks lead to this trivial case (as opposed to the alternative where X is non-zero and all the tokens are identical non-zero)?
2. Does theorem 2/corollary 1 hold for non-orthogonal value weights? For example other random matrices like iid Gaussian/uniform should all give the same cosine similarity properties as orthogonal in the large-width limit.
3. The LayerNorm without centering on line 128 is exactly RMSNorm https://arxiv.org/abs/1910.07467 which is popularly used in LLMs like LLama or Mistral instead of LayerNorm.
4. Why does this work only provide exponential convergence rates whereas the original rank collapse paper has doubly exponential?

Typo:
1. $d_N$ not $d_d$ in line 128.

**Limitations:**

There is not a clear limitations section but the authors do mention a limitation on line 283.

---

> ### Author Rebuttal · Authors · 2024-08-07
>
> We greatly appreciate your positive assessment and insightful comments, which have helped strengthen our work. Below, we provide individual responses the comments you raised.
>
> **Weaknesses**
>
> **W1** We agree with the reviewer that one should be aware of the effect of skip connections when analyzing rank collapse in transformers. Because we were mainly motivated by analyzing the sole effect of attention masks and LayerNorm, we excluded skip connections in our analysis for a controlled study. Building on our work, future research could include analyzing the effects of skip connections as well as positional encoding, among others.
>
> In the direction of the comment, we have conducted preliminary experiments measuring the evolution of $\mu(X)$ in SANs with skip connections alone and with both LayerNorm and skip connections. The results (Fig. 1 & 2 in the rebuttal) offer interesting insights:
>
> - For both initialized and pretrained models, adding pure skip connections indeed prevents $\mu$ from converging to 0, confirming existing theory. However, it seems to make $\mu$ unstable, particularly in deeper layers (Fig. 2). Compared with full transformers where $\mu$ stays relatively stable, there is a clear discrepancy.
>
> - Incorporating LayerNorm seems to alleviate this stability issue:
>     - In initialized models, LayerNorm alone effectively prevents rank collapse without causing $\mu$ to diverge.
>     - In pretrained models, both LayerNorm and skip connections mitigate rank collapse while LayerNorm is key to maintaining tokens at a scale consistent with full transformers, as we discussed in line 265 of the paper.
>
> These findings underscore the complex interplay between different components in transformers. LayerNorm emerges as a crucial element in mitigating rank collapse and stabilizing token representations while also counteracting potential negative effects of pure skip connections. We plan to further explore this in theory in the future.
>
> **W2** After carefully checking the code, we found that we accidentally loaded the results for initialized models when plotting for the pretrained models. We feel indebted to the reviewer for spotting this mistake. The results for pretrained models in Fig. 2 & 3 are now reported in Fig. 1 & 3 of the rebuttal, respectively.
>
> - For the effect of the temperature, the error bars are indeed very narrow for initialized models. One hypothesis for the effect to disappear after 5 or 6 layers at initialization is that in deeper layers as rank collapse happens, tokens align with each other and the temperature naturally has much less effect (especially given random $W_K$ and $W_Q$ cannot account for the increased similarities between tokens). We leave the thorough investigation for future work. As for pretrained models, the effect of temperature now can be observed in all layers and it is significant for all masks.
>
> - Due to the space limit, the results for 128 layer initialized models were provided in Appendix G.1. They exhibit the same trend as 12 layer models. We have also included 128 layer versions of Fig. 2 & 3 for initialized models over 3000 samples in the rebuttal.
>
> - For pretrained SANs, we follow Dong et al. (their official implementation: https://github.com/twistedcubic/attention-rank-collapse) by taking the existing pretrained models and only considering the self-attention layers.
>
> **W3** Dong et al. claims that self-attention with LayerNorm *cannot* prevent rank collapse. These counterexamples are sufficient to show that self-attention with LayerNorm *can* prevent it from happening. We do not claim that they are the only possible weights that work. It merely establishes that the set of such desirable designs is not empty.
>
> Fig. 1 in the rebuttal shows that just adding LayerNorm to SANs (no any other components such as skip connections) prevents rank collapse in initialized models, while in pretrained models LayerNorm mitigates the issue together with other modules and helps token representations maintain a scale consistent to full transformers. This empirically validates our theory.
>
> **W4** We appreciate this detailed feedback. We would like to clarify a misunderstanding here:
>
> - The counterexample works not because of zero activations in the center node (token 1 in this case). *Even with zero activations in the center node, rank collapse can still occur*:
>
>     - In Section 4.3.1, when token 2 is initialized on the olive segment in Fig. 1 of the paper, rank collapse still happens.
>     - In the case without LayerNorm, suppose we have the causal mask, and $X_{1,:}^{(0)}$ is a vector with a zero in its elements, and all $W_V^{(t)}$ are identity with $W_K^{(t)}$ and $W_Q^{(t)}$ satisfying A2). Then token 1 would stay at its initial value (so there is a deliberate zero activation from the start) and this would be the case where “attention allows other tokens to see the center node's activations but the zero activations are not being transferred to other tokens” --- yet rank collapse would still happen here (Theorem 1).
>
> - *The right intuition here can be better elucidated from a dynamical system perspective.* Without LayerNorm, rank collapse happens as the center node attracts all tokens to align. In the counterexample, the key insight to prevent token 2 from aligning with token 1 (assuming token 1 has converged to $v_1$) is that token 2's updated representation must cancel token 1's attraction, i.e. $X^{(t)}\_{2,:}W$ needs to have a component negating $v_1$. In the example, $W$ satisfies $Wv_2 = v_1+v_2$ for some $v_2$, so a $-v_2$ component in $X^{(t)}_{2,:}$ can negate token 1's effect ($v_1$). The crucial role of LayerNorm here is to stabilize this process by readjusting token scales to ensure that the cancellation still persists in subsequent updates. This key scaling effect of LayerNorm can also be observed from the newly added experiments, as we discussed in W1.
>
> ----
> **Due to the character limit, we answer the reviewer's questions in a comment below.**

---

> ### Author Response · Authors · 2024-08-07
> **Answers to the questions raised by Reviewer 7spk**
>
> **Questions**
>
> > Q1: The definition of mu(X) doesn't rule out that simply X converges to 0 in Frobenius norm, so that the activations simply go to 0, as opposed to saying something about the different tokens becoming aligned (in terms of cosine similarity going to 1) which is what some previous works have described as rank collapse. Do different attention masks lead to this trivial case (as opposed to the alternative where X is non-zero and all the tokens are identical non-zero)?
>
> Thank you for the question.
>
> - In the case with LayerNorm, X would not converge to 0 and hence the convergence of $\mu(X)$ to zero implies convergence to the same point on the unit sphere. In this case, it is equivalent to the cosine similarity going to one.
>
> - Without LayerNorm, whether or not $X$ goes to zero as $\mu$ goes to zero depends on many factors, including the mask, the initial inputs, and the model parameters.
>
>
>
> > Q2: Does Theorem 2/Corollary 1 hold for non-orthogonal value weights? For example other random matrices like iid Gaussian/uniform should all give the same cosine similarity properties as orthogonal in the large-width limit.
>
> This is a good point. We believe that Theorem 2 and Corollary 1 could be extended to more general classes of matrices, such as families of random matrices, as the reviewer mentioned. We leave this for future work.
>
> > Q3: The LayerNorm without centering on line 128 is exactly RMSNorm which is popularly used in LLMs like LLama or Mistral instead of LayerNorm.
>
> Thank you for the reference! This is a good call. We were aware of RMSNorm but instead chose the name “LayerNorm” following the convention of previous works such as Geshkovski et al. 2023 and Tian et al. 2023 to avoid confusion of terminology. We will add a footnote for this point.
>
> > Q4: Why does this work only provide exponential convergence rates whereas the original rank collapse paper has doubly exponential?
>
> This is a good question. Our result here is actually tight because in general for linear systems (which is the special case of $W^{(t)}_K=W^{(t)}_Q=0$, $W_V^{(t)}$ being the identity matrix and no LayerNorm), one can never have a doubly exponential convergence (see Antsaklis and Michel, Section 4.8). For this reason, we believe that the results in Dong et al. 2020 are specific to their setting and would not trivially extend to other settings, including ours.
>
>
> We appreciate your questions and comments very much. Please let us know if there are any further questions.
>
> ----------------------------------
> **References**
>
>
> Dong et al. Attention is not all you need: pure attention loses rank doubly exponentially with depth. In ICML, 2020.
>
> Geshkovski et al. A mathematical perspective on transformers. ArXiv, abs/2312.10794, 2023.
>
> Tian et al. Scan and snap: Understanding training dynamics and token composition in 1-layer transformer. In NeurIPS, 2023.
>
> Antsaklis and Michel. A Linear Systems Primer. 2000.

---

> > ### Comment · Reviewer_7spk · 2024-08-08
> > **Additional Questions**
> >
> > Thank you for the rebuttal and clarifications. I have some additional questions/comments in light of the response:
> >
> > 1. Doesn't the SANs+LN at initialisation plot (in Figure 1 and Figure 2 of additional pdf) contradict Theorem 2, which states that even with LN and with orthgonal initialisation you should obtain rank collapse? How are the QKV weights initialised in the plots? The mechanism outlined in W4 for the counterexample, while interesting, seems to rely on specific properties of the value weights/representation X which won't occur at random initialisation.
> > 2. Is it necessary to have certain neurons in the activations to have zero values in order to construct the counterexample? If not, it feels like an unnecessary detail which complicates the intuitive picture. The intuition of W4 should be included in the main paper to help readers.
> > 3. Regarding W1, a large body of existing work has shown that the combination of Pre-Norm skip connections reduces the effect of the residual branch which mitigates signal propagation degeneracies like rank collapse. Perhaps the main citation for this is https://arxiv.org/abs/2002.10444, but see also https://arxiv.org/abs/2010.12859 https://arxiv.org/abs/2003.04887  or https://arxiv.org/abs/2102.06171 to name a few. In the context of transformers, Noci et al 2022 and also https://arxiv.org/abs/2311.01906 have shown this effect too. I would recommend including the new plots and a discussion of these more practical architectures in the paper.
> > 4. I still maintain that the fact that the definition of mu(X) doesn't separate out the cosine similarities going to 1 vs the activation norms going to 0 as problematic for the understanding of the mechanisms of these different architectural components on rank collapse. Even if theoretically it is not possible to show, I would devise separate metrics to isolate these two effects and produce the equivalent of Figures 1 and 2.

---

> ### Author Response · Authors · 2024-08-09
> **Answers to the additional questions**
>
> We thank the reviewer for the quick response and the additional questions. Below, we provide point-to-point answers:
>
> **Q1**: Thank you for the comment.
>
> - QKV are initialized using the $U(-\sqrt{k}, \sqrt{k})$, with $k$=1/input_dim, which is the default initialization used in transformers like BERT implemented in HuggingFace.
>
> - Theorem 2 is established under the assumption that the value matrices are orthogonal. It is important to note that while finite random uniform matrices are orthogonal in expectation, each realization is *not* orthogonal. Hence the phenomenon in experiments does not contradict our theoretical results.
>
> - Note that due to the existence of LayerNorm, token trajectories are not a continuous function of model parameters, i.e. the value matrices. To get an idea, notice that if $x_1, x_2$ go to zero, the normalized values $x_1/||x_1||_2$ and $x_2/||x_2||_2$ can have a distance as large as two. As a result, any measure on token trajectories, including $\mu$, is not a continuous function of value matrices.
>
>
> **Q2**: Thank you for the question. As detailed in the response to W4 with concrete examples, having zero activations in the center node does *not* play any role in the mechanism of counterexample. If one replaces $W$ in the example with $W_{Z} = Z^{T}WZ$ where $Z$ is an orthogonal matrix, then the trajectory of $X^{(t)}$ of the new dynamics would be $X_Z^{(t)} = X^{(t)}Z$. Notice that under the new dynamics, the first token is going to converge to $(0,1)Z$, which clearly may not have any zero activation.
>
> This is a good point. We will add a remark to include a detailed discussion of the right intuition as the one provided to W4 to avoid confusion and misinterpretation. Thank you for bringing this up.
>
> **Q3**: Thank you for the comment and the pointers to the references. We are glad that the reviewer found our newly added experiments meaningful and connected to the literature. We will include the newly added experiments in the updated version of the manuscript and discuss the complex interplay between different architectural components in transformers and the related literature.
>
> **Q4**: Thank you for the comment. Our definition of $\mu$ is standard: it is mathematically equivalent to the definition of the measure $\textbf{res}$ used in Dong et al. (the original rank collapse paper), and the definition adopted in more recent e.g. Geshkovski et al. to study convergence of tokens in transformers. This measure checks for whether all tokens converge to the same representation or not. While it does not give information about whether the common representation is zero or not, in both cases the model loses representation power as it can no longer map tokens within the same sequence to different values.
>
> We appreciate your questions and comments very much. Please let us know for any further questions.

---

> > ### Comment · Reviewer_7spk · 2024-08-10
> > **Thank you**
> >
> > Thanks for the response. Regarding Q1, it would be good to do the relevant plot at initialisation matching the assumptions of Theorem 2 (e.g. orthogonal weights) just as a verification of the theory. I am surpised that SAN+LN does not converge to rank collapse at initialisation, and I remain unconvinced that the theoretical counterexamples provided explain this because I don't think they should not apply at initialisation. But otherwise thanks again for the clarifications.

---

> > > ### Author Response · Authors · 2024-08-12
> > > **Response to the comments**
> > >
> > > We thank the reviewer for the comment.
> > >
> > > - To further address the concern regarding the verification of our Theorem 2, we perform an additional set of experiments of SAN+LN under the exact conditions in Theorem 2 with initialization of the value matrices set to be exactly orthogonal. Here is the table summarizing the mean (std) of $\mu$’s in a 128 layer networks under different masks:
> > >
> > > | layer  |  complete           | causal            | slide-window      | slide-window-uni  |
> > > |-----|--------------------|-------------------|-------------------|-------------------|
> > > | 0   | 274.7604 (2.0498)  | 274.7832 (2.0682) | 274.6891 (1.9408) | 274.7870 (2.0094) |
> > > | 32  | 4.6109e-5 (2.0682) | 47.1223 (16.2438) | 111.6686 (3.4742) | 161.0730 (2.5183) |
> > > | 64  | 4.6347e-5 (2e-6)   | 67.9591 (12.3382) | 95.4022 (4.8795)  | 160.5059 (3.8947) |
> > > | 96  | 4.6396e-5 (2e-6)   | 82.9471 (11.5331) | 85.6265 (5.5486)  | 164.1741 (5.2969) |
> > > | 128 | 4.6220e-5 (2e-6)   | 92.3288 (11.6265) | 79.9934 (5.6181)  | 165.205 (5.4153)  |
> > >
> > > In particular, we observe that while $\mu$ converges to zero for complete masks and shows a clear decreasing trend for slide-window masks (both masks are strongly connected), such converging trends do not hold for non-strongly connected masks (causal and sliding-window-uni, as they are only quasi-strongly connected), verifying our Theorem 2 and showing that (1) the convergence rate depends inversely on the graph diameter (2) the tightness of our result that Theorem 2 only applies for strongly connected masks. We will include these sets of experiments to better illustrate our theoretical results.
> > >
> > > - Finally, we would like to emphasize again that the goal of the counterexample is to establish that self-attention with layer normalization can prevent rank collapse from happening, as opposed to what Dong et al. claims (“layer normalization plays no role”). We do not claim that they are the only possible weights that can work. It merely establishes that the set of such desirable designs exists.
> > >
> > >
> > > We thank the reviewer once again for the discussion and the constructive feedback!

---

> > > > ### Comment · Reviewer_7spk · 2024-08-13
> > > > **Thank you**
> > > >
> > > > Thank you for the additional clarifications and empirical verification of Theorem 2. I agree that including these results as well as some of the other suggestions (like the updated figures in the rebuttal pdf, clarifying the intuition for the counterexamples, or discussion of omitted architectural components like skips/positional encodings) will strengthen the paper. I would also mention the goal of the counterexample in order to better distinguish when these theoretical results are expected to hold in practice (and in the empirical verifications), for a cleaner message.
> > > >
> > > > I think with these updates the paper would make a nice contribution to NeurIPS 2024. I will keep my score of 6.

---

> > > > > ### Author Response · Authors · 2024-08-13
> > > > > **Thank you**
> > > > >
> > > > > We are happy to read this sentiment and will diligently incorporate the updates in the camera-ready version of our paper. We sincerely thank you for your valuable input!

---

### Official Review · Reviewer_yhKR · 2024-07-15

**Soundness:** 3
**Presentation:** 3
**Contribution:** 3
**Rating:** 7
**Confidence:** 3

**Summary:**

In this work, the authors investigate the issue of rank collapse in Transformers, providing insights into how attention masks and layer normalization can mitigate this problem. The paper includes extensive analysis, addressing two important questions and offering valuable contributions to the field.

**Strengths:**

1.I like this paper for its strong motivation and very interesting insights. I appreciate the authors for addressing the rank collapse issue in Transformers, a topic often overlooked in the community.

2. The authors provide detailed analysis to demonstrate that attention masks and layer normalization can help address this issue.

3. The experimental results support the authors' analysis effectively.

4. Figure 1 is particularly helpful in understanding the illustration of the effectiveness of layer normalization.

**Weaknesses:**

It seems obvious that causal masking and local attention would help mitigate the issue of rank collapse in Transformer/Attention mechanisms compared to full attention. Maybe no need to demonstrate this a lot.

**Questions:**

I like this work; however, I have two additional questions:

If local attention alleviates rank collapse, how do the experimental results compare to those of full attention or causal masked attention?

Which approach is more effective in addressing the rank collapse issue: local attention or causal attention? Are there any strong insights into this? I assume local attention may not be generally applicable in language tasks but is more suitable for vision tasks.

**Limitations:**

Please see above.

---

> ### Author Rebuttal · Authors · 2024-08-07
>
> We appreciate your thoughtful comments and positive assessment of our work. After carefully reviewing your feedback, below we provide answers to the comments you raised.
>
>
> > Q1: It seems obvious that causal masking and local attention would help mitigate the issue of rank collapse in Transformer/Attention mechanisms compared to full attention.
>
> Thank you for the comment. While the conclusion might seem intuitive, we would like to point out that formalizing it in theory with rigorous mathematical proofs turns out to be very nontrivial — causal masking has been populated by GPTs for years, yet none of the existing theoretical works on analyzing rank collapse [1,2,3,4] in transformers can accommodate transformers with causal masking or local attention. As we discussed in the paper, all those works require having full attention as a *necessary* assumption to derive their theoretical results on rank collapse. To tackle this technical difficulty, we take a novel graph-theoretic approach by formalizing causal masking and local attention through the lens of directed graphs and make use of both graph theory and discrete dynamical systems theory tools. Moreover, our results establish a formal connection between the rate of rank collapse and the graph structure. We hope this novel analysis technique would be a useful tool for the literature and future research in the community.
>
> > Q2: If local attention alleviates rank collapse, how do the experimental results compare to those of full attention or causal masked attention?
>
> The experimental results can be found in Figure 1, left column. We choose sliding window attention (SWA) deployed in Longformer and more recently Mistral 7B as the representative for local attention. Compared with both full and causal attention, the convergence is clearly slower, which confirms our theoretical results.
>
> > Q3: Which approach is more effective in addressing the rank collapse issue: local attention or causal attention? Are there any strong insights into this? I assume local attention may not be generally applicable in language tasks but is more suitable for vision tasks.
>
> For pure self-attention networks, local attention would be more effective for mitigating rank collapse. Our theory suggests that the rate of rank collapse is directly affected by the diameter of the directed graph (Theorem 1), which is confirmed with numerical experiments (Figure 1).
>
> Regarding the use of local attention, it is indeed popular in vision tasks. See [5,6,7] for references. However, due to the demand for efficiency in long-context, local attention is also getting popular in language tasks. For example, local and sparse attention was populated by Longformer [8] and OpenAI [9] and nowadays LLMs like Mistral 7B use sliding window attention (SWA) [10]. Other popular sparse attention for language tasks include, but not limited to, BigBird [11], Recurrent Memory Transformers (RMTs) [12,13] and StreamingLLM [14].
>
> We appreciate your questions and comments very much. Please let us know for any further questions.
>
> ----------------
> **References**
>
> [1] Dong et al. Attention is not all you need: pure attention loses rank doubly exponentially with depth. In ICML, 2020.
>
> [2] Noci et al. Signal propagation in transformers: Theoretical perspectives and the role of rank collapse. In NeurIPS, 2022.
>
> [3] Geshkovski et al. A mathematical perspective on transformers. 2023.
>
> [4] Geshkovski et al. The emergence of clusters in self-attention dynamics. In NeurIPS, 2023.
>
> [5] Liu et al. Swin transformer: Hierarchical vision transformer using shifted windows. In ICCV, 2021.
>
> [6] Pan et al. Slide-Transformer: Hierarchical Vision Transformer with Local Self-Attention. In CVPR, 2023.
>
> [7] Hassani et al. Neighborhood Attention Transformer. In CVPR, 2023.
>
> [8] Beltagy et al. Longformer: The long-document transformer. 2020.
>
> [9] Child et al. Generating Long Sequences with Sparse Transformers. 2019.
>
> [10] Jiang et al. Mistral 7B. 2023.
>
> [11] Zaheer et al. Bigbird: Transformers for longer sequences. In NeurIPS, 2020.
>
> [12] Bulatov et al. Recurrent Memory Transformer. In NeurIPS, 2022.
>
> [13] Bulatov et al. Scaling Transformer to 1M tokens and beyond with RMT. 2024.
>
> [14] Xiao et al. Efficient Streaming Language Models with Attention Sinks. In ICLR, 2024.

---

### Author Rebuttal · Authors · 2024-08-07

## Response to all reviewers


We would like to thank the reviewers for carefully reading our paper and giving insightful comments and constructive feedback.
We are glad that our work was recognized as “interesting”, “rigorous” (Reviewer ivrv) and “offering valuable contributions to the field” (Reviewer yhKR), including “a graph-theoretic approach to analyze self-attention” (Reviewer ivrv) and “new insights into the rank collapse phenomenon” (Reviewer 7spk). We are also encouraged that the reviewers found our paper "well-written" (Reviewer 7spk).

We have provided detailed responses to each of the reviews separately. We also include additional numerical results in the rebuttal pdf under the same experimental setup as in our paper. Specifically, we increase the sample size of the experiments to 3000 in all experiments and consider the setting of 128-layer randomly initialized models when measuring the effects of different attention masks, LayerNorm and the temperature term.

The authors.

---

### Decision · Program_Chairs · 2024-09-25

**Decision:**

Accept (poster)

**Comment:**

This paper studies the rank collapse problem of attention, focusing on two previously ignored aspects: attention masks and layer normalization. Being a theoretical paper, the authors did a good job providing rigorous analyses to these two important design choices which I believe could help further the community's understanding of attention and Transformers. There are concerns raised, eg on the practical implications of this work -- I believe these are nice to have but shouldn't be the blocker for accepting a theoretical work like this. I'm leaning towards accepting this work and encourage the authors to update the paper with materials presented in the rebuttal period.